



# The preserved plume of the Caribbean Large Igneous Plateau revealed by 3D data-integrative models

Ángela María Gómez-García[1,2,3], Eline Le Breton[4], Magdalena Scheck-Wenderoth[1], Gaspar Monsalve[2], Denis Anikiev[1]

[1]GFZ German Research Centre for Geosciences, Potsdam, 14473, Germany.
[2]Universidad Nacional de Colombia – Medellín. Facultad de Minas.
[3]CEMarin – Corporation Center of Excellence in Marine Sciences.
[4]Institute of Geological Sciences, Freie Universität Berlin, Berlin, 12249, Germany.

*Correspondence to*: Ángela María Gómez-García (angela@gfz-potsdam.de)

**Abstract.**

Remnants of the Caribbean Large Igneous Plateau (CLIP) are found as thicker than normal oceanic crust in the Caribbean Sea, that formed during rapid pulses of magmatic activity at ~91–88 Ma and ~76 Ma. Strong geochemical evidence supports the hypothesis that the CLIP formed due to melting of the plume head of the Galápagos hotspot, which interacted with the Farallon (Proto-Caribbean) plate in the east Pacific. Considering the plate tectonics theory, it is expected that the lithospheric portion of the plume-related material migrated within the Proto-Caribbean plate, in a north - north-eastward direction, developing the present-day Caribbean plate. In this research, we used 3D lithospheric-scale, data-integrative models of the current Caribbean plate setting to reveal, for the first time, the presence of positive density anomalies in the uppermost lithospheric mantle. These models are based on the integration of up-to-date geophysical datasets, from the Earth's surface down to 200 km depth, which are validated using high-resolution free-air gravity measurements. Based on the gravity residuals (modelled minus observed gravity), we derive density heterogeneities both in the crystalline crust and the uppermost oceanic mantle (< 50 km). Our results reveal the presence of two positive mantle density anomalies beneath the Colombian and the Venezuelan basins, interpreted as the preserved fossil plume conduits associated with the CLIP formation. Such mantle bodies have never been identified before, but a positive density trend is also indicated by S-wave tomography, at least down to 75 km depth. The interpreted plume conduits spatially correlate with the thinner crustal regions present in both basins; therefore, we to propose a modification to the commonly accepted tectonic model of the Caribbean, suggesting that the thinner domains correspond to the centres of uplift due to the income of the hot, buoyant plume head. Finally, using six different kinematic models, we test the hypothesis that the CLIP originated above the Galápagos hotspot; however, misfits of up to ~3000 km are found between the present-day hotspot location and the mantle anomalies, reconstructed back to 90 Ma. Therefore, we shed light on possible sources of error responsible for this offset and discuss two possible interpretations: (1) The Galápagos hotspot migrated (~1200-3000 km) westward while the Caribbean plate moved to the north, or (2) The CLIP was formed by a different plume, which – if considered fixed - would be nowadays located below the South American continent.

## 1 Introduction

Oceanic plateaus are vast areas characterised by a thicker than "normal" oceanic crust, which might reach up to 38 km (Kerr and Mahoney, 2007). Although about 12 different oceanic plateaus have been recognised worldwide, they represent one of the least well-known Earth's magmatic processes (Kerr, 2014). In the early 70's, Edgar et al. (1971) and Donnelly (1973) discovered the Caribbean Large Igneous Plateau, which corresponds to the second largest plateau (by area) after the Ontong Java, with an approximated extent of 1.1 x $10^6$ km$^2$, and an estimated excess magma volume of 4.4 x $10^6$ km$^3$ (Kerr, 2014).





The origin of such vast volume of basalt is widely recognised as the interaction of a mantle plume with the overriding, mobile

lithosphere. With time, the plume-related material suffers physical and chemical changes, which at the same time are associated with a diversification in the way the plume interacts with the overriding plate. During the lifespan of a plume it is expected to create first extensive oceanic plateaus, followed by aseismic ridges, due to melting of the large plume head or the narrower plume tail, respectively (Campbell, 2005). The initial stages of the plume-lithosphere interaction include the uplift and weakening of the overlying lithosphere, due to the income of hot, highly buoyant mantle material. At a later stage, when the

plume is no longer active, geodynamic models show that the frozen plume material can be preserved into the lithosphere, forming high-density, and therefore high-velocity bodies (François et al., 2018).

Successful detections of present-day mantle plumes using P-wave and/or S-wave velocity anomalies include the work of e.g. Montelli et al. (2004) and Civiero et al. (2019). These results suggest that the currently active plumes are characterised by negative velocity anomalies, associated with the presence of high-temperature material. The plume conducts show a variety

of shapes, some of them including the interconnection of branches at different depths (e.g. Civiero et al., 2019 and references therein). Imaging these complex systems, however, has implied big challenges in the scientific community, especially for the correct interpretation of tomographic images (Campbell, 2005; Civiero et al., 2019).

The oceanic plateaus are normally difficult to subduct due to their abnormal thickness and positive thermal imprint inherited from their mantle plume origin. Thus, fragments of the Caribbean plateau have been accreted along continental margins, such

as in Ecuador, Colombia, Panama, Costa Rica, Curacao and Hispaniola (Hastie and Kerr, 2010; Thompson et al., 2004). Using accreted material and relatively few drilled or dragged submarine rock samples, the geochemistry of the CLIP has been reconstructed (e.g. Geldmacher et al., 2003; Hastie and Kerr, 2010; Kerr and Tarney, 2005; Thompson et al., 2004). Indeed, strong geochemical evidence suggests that the CLIP corresponds to melting of the plume head of the Galápagos hotspot (Geldmacher et al., 2003; Thompson et al., 2004), although recent kinematic reconstructions of the Caribbean do not allow to

trace back the location of the plate above the present-day location of the Galápagos plume (Boschman et al., 2014). Nevertheless, diverse evidence exists about the north - north-eastward migration of the Caribbean plate from the east Pacific. The present-day Caribbean plate is composed of different accreted crustal domains (e.g. volcanic arcs, continental and oceanic realms) which have migrated since Late Jurassic to Early Cretaceous times, including the igneous plateau materials that affected the oceanic crust of the former Farallon plate (Boschman et al., 2014; Montes et al., 2019a).

Different regions of the Caribbean have been the target of relatively extensive seismic reflection and refraction, sonar, as well as drilling campaigns (e.g. Diebold and Driscoll, 1999; Edgar et al., 1971; Kroehler et al., 2011; Mauffret and Leroy, 1997; Rosencrantz, 1990), some of them undertaken with the limitations of the early seismic acquisition. Nonetheless, the coverage of these measurements is poor when compared with the complexity of most of the Caribbean morphological structures.

In this paper, the main goal is to evaluate the present-day 3D lithospheric structure of the South Caribbean margin (continental

and oceanic domains -black box in Fig. 1) by means of modelling of the gravity anomalies (Gz), which are especially sensitive to deep density distributions (Álvarez et al., 2014, 2015); and therefore, a potential tool for analysing the upper 200 km of the



Earth. Here, the high-spatial resolution EIGEN6C-4 dataset is used (Förste et al., 2014; Ince et al., 2019), which includes a spherical harmonic solution of up to degree and order Nmax=2190, equivalent to a topographic wavelength of ~18 km.

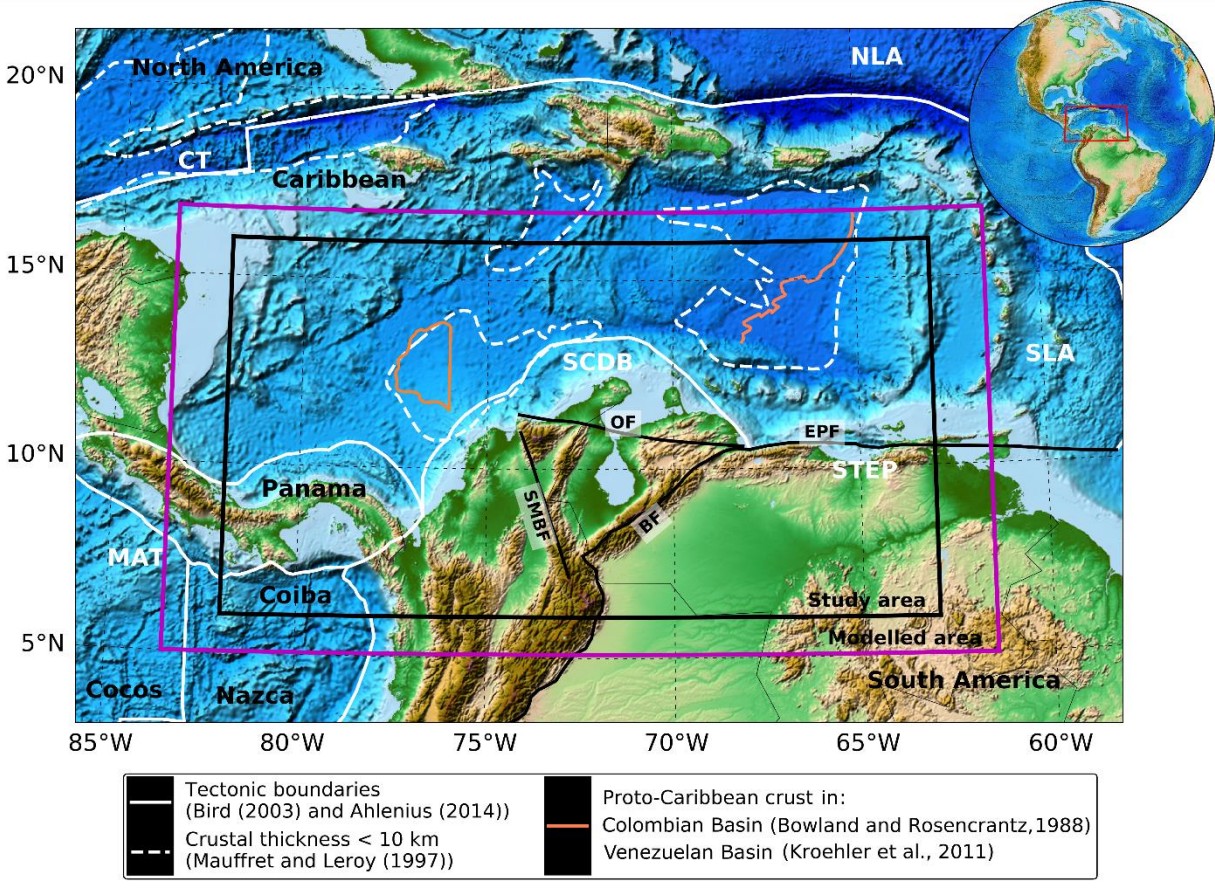

Figure 1: Location of the regions for models (magenta box) and interpretations (black box). The modelled area involves four tectonic plates: Caribbean, Panama, Nazca (Coiba microplate) and north of South America. Black lines represent main faults. BF = Boconó Fault, CB = Colombian Basin, CT = Cayman Trough, EPF = El Pilar Fault, NLA = North Lesser Antilles subduction, MAT = Middle American Trench, OF = Oca Fault, SCDB = South Caribbean Deformed Belt, SLA = South Lesser Antilles subduction, SMBF = Santa Marta-Bucaramanga Fault, and STEP = Subduction-Transform-Edge-Propagator fault system, VB = Venezuelan Basin. Shade relief image from Amante and Eakins (2009).

Due to the fact that the gravity response of a system is the superposition of the gravity effects caused by all the density contrasts within it, we considered the gravitational effects caused by the heterogeneous lithospheric mantle in the South Caribbean and north-western South American plates. Therefore, the geometries of both the Nazca and the Caribbean flat-slabs were included in the gravity models.

Previous studies in this region include few 3D lithospheric-scale, gravity-validated models (Gómez-García et al., 2019b; Sanchez-Rojas and Palma, 2014). However, some limitations of these attempts include, for instance, a spatially heterogeneous gravity dataset, a mantle considered as having a uniform and constant density, or the analysis of only the shallow density contrasts.



The results of the gravity inversion not only highlight crustal areas heavily affected by the high-density plume material, but
also suggest the presence of a high-density trend in the oceanic mantle of the Caribbean plate. This trend can be followed from
the Moho down to 75 km depth, as high S-wave velocities in the tomographic model SL2013sv support (Schaeffer and
Lebedev, 2013). These results are interpreted as the preserved, lithospheric fossil plume conduits, responsible for the
development of the CLIP, which migrated as the Proto-Caribbean lithosphere moved from the east Pacific.

Finally, taking advantage of the more precise spatial location of the CLIP fossil plume conduits, different kinematic
reconstructions are explored, aiming to evaluate the hypothesis that the CLIP formed above the Galápagos hotspot. However,
offsets of ~1200-3000 km are obtained between the present-day Galápagos hotspot location and the plume conduits back to
90 Ma, as previously reported by Pindell et al. (2006) and Boschman et al. (2014).

## 2 Tectonic setting of the Caribbean and north-western South America

### 2.1 The Caribbean Large Igneous Plateau

The Caribbean oceanic crust is the complex product of the interaction of a mantle plume with the Proto-Caribbean (Farallon)
plate 90 Ma ago, forming the second largest plateau after the Ontong Java (Kerr, 2014). Melt modelling of the plateau high
MgO lavas found in Curaçao suggest that the primary magmas contained up to 24wt% MgO, and that they correspond to a 30-
32% partial melting of a fertile, heterogeneous peridotite, with a potential temperature of 1564-1614 °C (Hastie and Kerr,
2010; Kerr, 2014).

Due to the fact that most of the plateau is currently submerged below large water depths, its structure and geochemistry has
been mainly constrained from accreted material along the Caribbean margins, including Colombia, Ecuador, Costa Rica,
Jamaica, and Hispaniola (Kerr, 2014; Kerr and Tarney, 2005; Van Der Lelij et al., 2010). In summary, the accreted fragments
consist of basaltic and picritic lavas and sills, with gabbros and ultramafic cumulates (Kerr, 2014).

Diebold and Driscoll (1999) and Driscoll and Diebold (1999) presented the most detailed model of the formation of the CLIP,
based on seismic reflection profiles. According to their model, the Proto-Caribbean crust formed in the east Pacific by seafloor
spreading in the Late Jurassic - Early Cretaceous time. This crust had a "normal" thickness (~6 km) that later on, during the
pre-Senonian, interacted with a mantle plume. As a result, a widespread eruption of the first basaltic flows, accompanied by
extension and thinning of the Proto-Caribbean crust took place. Additionally, the intrusion of igneous material and underplating
of residual mantle and ultramafic cumulates contributed to the formation of the thicker portions of the Caribbean plate, which
in some places can reach up to 20 km. Then, during the Senonian, additional extension and underplating occurred, causing the
uplift and rotation of the Beata Ridge, accompanied by the late stage of basaltic flows. Thus, at least two main pulses of
magmatic activity have been identified, at ~91–88 Ma and ~76 Ma (Diebold and Driscoll, 1999; Sinton et al., 1998).

Nowadays, it is possible to recognise anomalously thin, extended Proto-Caribbean crust, where crustal thickness ranges
between 2.8 and 5 km, in the south-eastern Venezuelan Basin and in some regions of the Colombian Basin. These thin areas
are characterised by a rough basement (B" horizon), suggesting that no basalt flows covered these domains. The Moho depth





also shows a spatial variation: it shallows abruptly below the rough B" basement, and deepens in areas where the B" horizon is smooth (Driscoll and Diebold, 1999). Figure 1 depicts the areas of thinned Proto-Caribbean crust as orange lines in the Colombian Basin (Bowland and Rosencrantz, 1988) (closed polygon) and in the Venezuelan Basin (Kroehler et al., 2011) (eastward of the orange line).

Although there is strong geochemical evidence that supports the origin of the CLIP as melting of the paleo-Galápagos plume head (Geldmacher et al., 2003; Hastie and Kerr, 2010; Kerr and Tarney, 2005; Thompson et al., 2004), recent kinematic reconstructions back to 90 Ma by Boschman et al. (2014) showed an offset of up to 3000 km between the present-day Galápagos hotspot and the location of the Caribbean plate.

## 2.2 Nazca and Caribbean subductions

The northern margin of the South American plate is an active zone with two flat-slab subductions which interact at depth: the Nazca (Coiba microplate) plate from the west and the Caribbean plate from the north (some recent references include: Bernal-Olaya et al., 2015; Chiarabba et al., 2015; Monsalve et al., 2019; Porritt et al., 2014; Siravo et al., 2019; Syracuse et al., 2016; Vargas and Mann, 2013; Wagner et al., 2017; Yarce et al., 2014). This complex interaction defines a poorly understood tectonic setting. Indeed, the geometry of these subductions is not well constrained yet, although different attempts have been made (e.g.

Van Benthem et al., 2013; Bezada et al., 2010; Hayes et al., 2018; Mora et al., 2017).

Since the Late Cretaceous, the oblique collision of the Caribbean plate due to its east-northeast migration from the Pacific Ocean has shaped the South Caribbean margin. As a consequence, several foreland basins have been formed in the north of the South American continent. They followed the diachronous displacement of the collision front, younging eastward from Eocene to Present time (Escalona and Mann, 2011 and references therein).

Along this margin, the Caribbean plate has subducted beneath the continental South American plate and the Maracaibo block since the Late Cretaceous (Kroehler et al., 2011), forming the South Caribbean Deformed Belt (SCDB –Fig. 1). However, the lateral extension of this subduction has been actively debated. For instance, the tomographic model of Van Benthem et al. (2013) showed positive P-wave velocity anomalies from 76° W to 67° W, only traced in the upper mantle, suggesting that the subduction is ongoing along most of the margin, but excluding the southwestern edges with the South American plate and the

Panama microplate. According to Kroehler et al. (2011), in the southeast portion of the Caribbean plate an incipient subduction might be developing nowadays.

From the Pacific, the subduction of the Nazca plate shapes the western margin of the South American continent. In Colombia, north of the Caldas tear (~5°N), the subduction becomes flat (Chiarabba et al., 2015; Vargas and Mann, 2013; Wagner et al., 2017) and is likely associated with the down going Coiba microplate (Fig. 1). This flat-slab subduction is characterised by a

buoyant oceanic crust, which Chiarabba et al. (2015) associated with a volcanic ridge.





## 3 Modelling approach

The first steps in the modelling workflow were to define a structural starting model and to assign the densities to the upper most lithospheric layers. Then, different mantle density configurations were tested (supplementary material S1). The gravity response of these 3D lithospheric-scale structural and density models were computed with the software IGMAS+ (Schmidt et
al., 2011), at 10 km calculation height. The modelled results were compared with the free-air gravity anomalies of EIGEN-6C4 (Förste et al., 2014), available at the Calculation Service of the International Centre for Global Earth Models –ICGEM (Ince et al., 2019). The modelled area is shown as a black box in Fig. 1, and ranges between approximately 5°-16° N and 62°-83° W.

In the next subsections, the datasets used to constrain the initial structural model are presented (Sect. 3.1). Additionally, because
the gravity anomalies of larger wavelengths are especially sensitive to deep density contrasts, the initial density configuration took into account the evaluation of the mantle density effect in the gravity model (Sect. 3.2). Finally, Sect. 3.3 provides the details about the forward modelling of the gravity residuals of the initial lithospheric configuration, which includes the definition of two high-density bodies in the uppermost lithospheric mantle (< 50 km). The general methodological workflow is shown in Fig. 2.

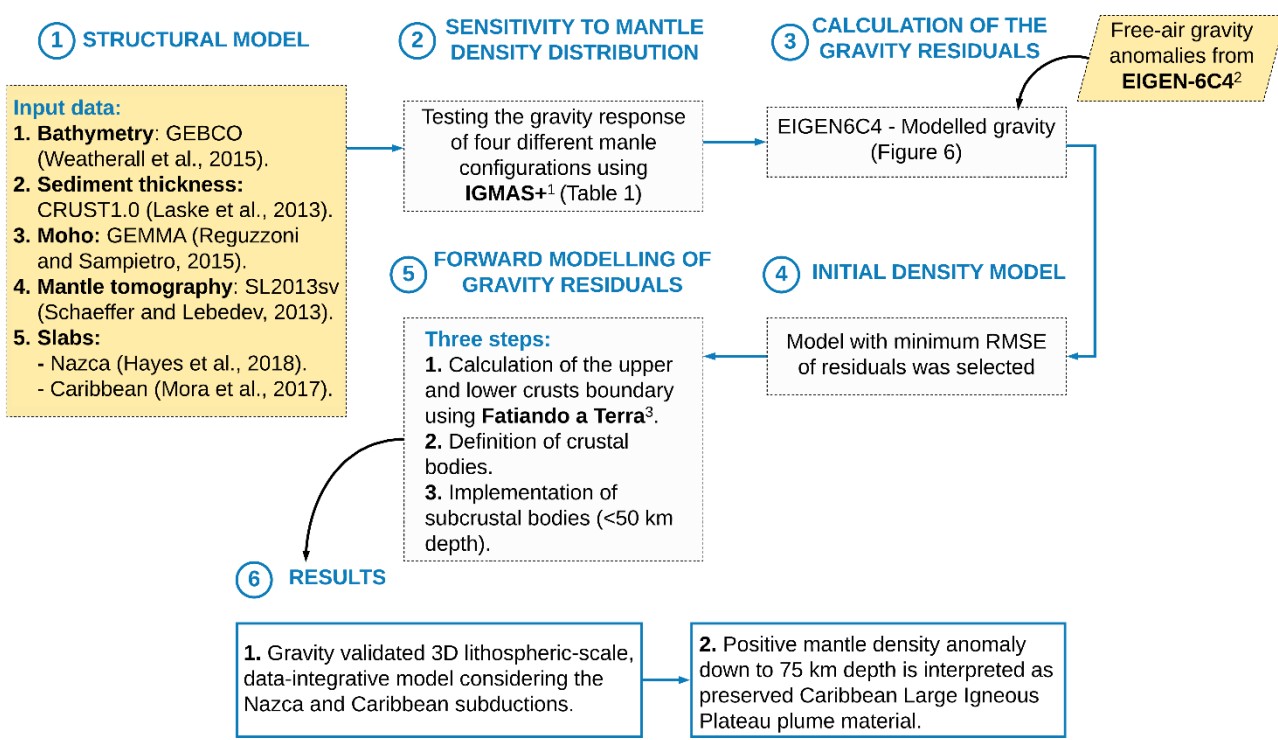


**Figure 2: Methodological scheme used in this research. [1] Schmidt et al. (2011). [2] Förste et al. (2014) and Ince et al. (2019). [3] Uieda et al. (2013).**



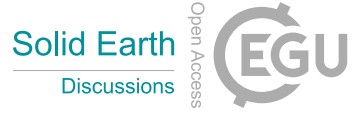

### 3.1 Input data

The lithospheric structural starting model includes the thicknesses of eight interfaces: (1) seawater (Fig. 3 (a)), obtained as the
difference between sea level and the General Bathymetric Chart of the Oceans (GEBCO) (Weatherall et al., 2015); (2) oceanic
and (3) continental sediments (Fig. 3 (b) and (c)) taken from the CRUST1.0 dataset (Laske et al., 2013); (4) oceanic and (5)
continental crystalline crust (Fig. 3 (d) and (e)) considering the Moho depth from the GEMMA model (Reguzzoni and
Sampietro, 2015); (6) the slabs shapes of the Nazca (Hayes et al., 2018) and (7) Caribbean (Mora et al., 2017) flat-slabs
subductions (Fig. 3 (f)); and finally, (8) the lithospheric mantle, which was subdivided into seven layers using the SL2013sv
S-wave velocity model (Schaeffer and Lebedev, 2013), from 50 to 200 km depth. The integration of the different datasets was
made after interpolating to a homogeneous spatial resolution of 25 km, using the minimum curvature algorithm with the
software Petrel (Schlumberger, 2019).

The sediment thickness from CRUST1.0 was selected not only because it includes the continental regions, but also because it
takes into account information from broadly distributed seismic profiles of EXXON (1985) that allow the recognition of the
main sedimentary features, without disregarding the spatial connection between both inland and offshore sedimentary systems.
The sea level was used as the boundary zone between the oceanic and the continental depocenters.

Additionally, the continent-ocean transition (COT) defined by Gómez-García et al. (2019) was used to separate both oceanic
and continental crustal types. The black dashed-lines in the oceanic crust map (Fig. 3 (d)) depict regions where the crust has a
thickness smaller than 10 km, according to Mauffret and Leroy (1997). Most of these regions coincide with places where the
calculated crustal thickness is extremely thin, with values lower than 4 km.

Finally, even though the 3D mantle densities published by Gómez-García et al. (2019a), (2019b) were considered (see Sect.
3.2), whose approach takes into account the mineralogical composition of the mantle and the S-wave velocities of Schaeffer
and Lebedev (2013), the gravity signal of the Nazca and Caribbean slabs was also tested. The Nazca subduction was taken
from the Slab2 dataset (Hayes et al., 2018), which integrates diverse geophysical information in the definition of the slab's 3D
structure, including: active-source seismic data, receiver functions, seismicity catalogues from local and regional networks and
tomographic models. On the other hand, the Caribbean slab was defined following the hypocentral distribution (Mora et al.,
2017). Figure 3 (f) presents the integrated thickness of both flat-slabs. In the region where both subducting plates interact at
depth the integrated thickness reaches more than 120 km. While the Slab2 dataset includes the Nazca slab thickness, in the
Caribbean case a thickness of ~70 km was a priori assumed (see Fig. S2).




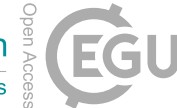


**Figure 3: Structural layers used in the starting model. (a) Water thickness from GEBCO (Weatherall et al., 2015). (b) and (c) unconsolidated oceanic and continental sediments based on CRUST1.0 (Laske et al., 2013). (d) and (e) oceanic and continental crystalline crust thicknesses. The dashed lines in panel (d) represent places where the oceanic crust is thinner than 10 km (Mauffret and Leroy, 1997). (f) Integrated slab thickness from Nazca (Hayes et al., 2018) and Caribbean (Mora et al., 2017) subductions. The independent thickness maps of the slabs are presented in Fig. S2. BAB = Barinas-Apure basin. FB = Falcón basin. LMB = Lower Magdalena basin. PDB = Panamá Deformed Belt. SCDB = South Caribbean Deformed Belt.**

## 3.2 Initial lithospheric configuration and the mantle density effect on the gravity signal

The lithospheric mantle density is one of the most influential parameters in the modelling of the gravity anomalies, because of its massive volume compared to the other lithospheric layers. The mantle density effect on the gravity signal was explored testing four different mantle configurations (see supplementary material S1). The model with the minimum RMSE compared with EIGEN-6C4 was selected as the initial lithospheric configuration. This model includes the 3D mantle densities obtained from the conversion of the S-wave velocity anomalies of the SL2013sv model (Schaeffer and Lebedev, 2013), from 50 to 200 km depth, published by Gómez-García et al. (2019a), (2019b). In this approach, the density conversion is performed following





a modified version of Goes et al. (2000) method, using a pressure and temperature dependent expansion coefficient (Hacker and Abers, 2004; Meeßen, 2017).

Figures S3 and S4 present the S-wave velocities at different depths (from 50 to 200 km, every 25 km) and their associated densities, respectively. In general, with this approach it is possible to establish a relation between high velocity anomalies and high mantle densities (blue and dark blue regions in Fig. S3 and S4, respectively), and vice versa with the low velocities and

low mantle densities. Figure 4 summarises the conversion results as described hereafter.

The mantle structure shows two different trends. At depths shallower than 75 km, in the Caribbean Sea the mantle is denser compared with the one below the South American continent, with differences up to 40 kg m$^{-3}$ at 50 km depth. Although the mantle density increases with depth, it is possible to distinguish a subduction-like behaviour from 125 to 200 km below the surface: denser (colder) material below the South American continent, in contrast with a lighter (warmer) mantle below the

ocean. These two trends are summarised in Fig. 4 (a) and (c) for depths shallower than 75 km, and in Fig. 4 (b) and (d) for those deeper than 125 km depth.

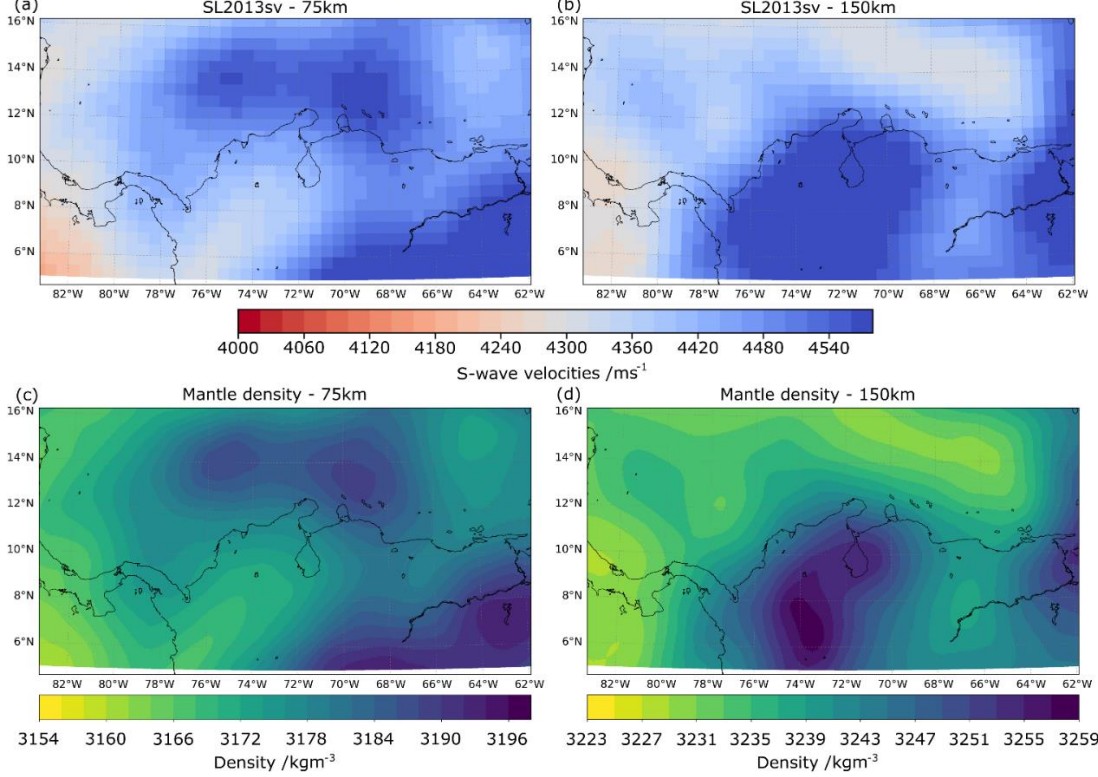

**Figure 4: Relation between the S-wave velocities from the SL2013sv model (Schaeffer and Lebedev, 2013) and the calculated mantle densities. (a) and (b) S-wave velocities at 75 and 150 km depth, respectively. The original cell size of the tomographic data is 0.5°x0.5°.**
**(c) and (d) corresponding densities at 75 and 150 km depth after Goes et al. (2000) and Meeßen (2017) (see text for details of the conversion). In this case the S-wave velocities were interpolated to a spatial resolution of 25 km to make them consistent with the spatial resolution of the rest of the lithospheric layers.**

Based on the spatial distribution of both parameters (velocity and density), it is not easy to differentiate which portions of the mantle belong to either the Caribbean or the Nazca slabs. Nevertheless, the fastest velocities (up to 4580 m s$^{-1}$) are concentrated

between 68° and 77° W, from 125 down to 200 km depth (Fig. 4 (b) and S3).

Because at depths shallower than 125 km the observed patterns do not allow to make a reliable interpretation about the slabs, it was necessary to explore the gravity effect of both subducting slabs using additional datasets. Figure 5 (a) shows the spatial distribution of the depth to the top of the Nazca (Coiba) slab, as published by Hayes et al. (2018). Two main factors can be highlighted from this geometry: (1) even though it has been published as the Nazca subduction, it is very likely that it also

includes an important fraction of the Caribbean slab, because of its northward extension. And (2) it is possible to distinguish its flat-shape, which widens towards the east, in the region north of the Caldas tear (~5° N), as previously reported by other authors (e.g. Chiarabba et al., 2015; Vargas and Mann, 2013). Similarly, Fig. 5 (b) depicts the depth to the top of the Caribbean slab by Mora et al. (2017), which according to these authors is present westward of ~72° W. This geometry suggests that the Caribbean subduction is not entirely flat, but a sharp increase on its angle is present towards the south and south-east.

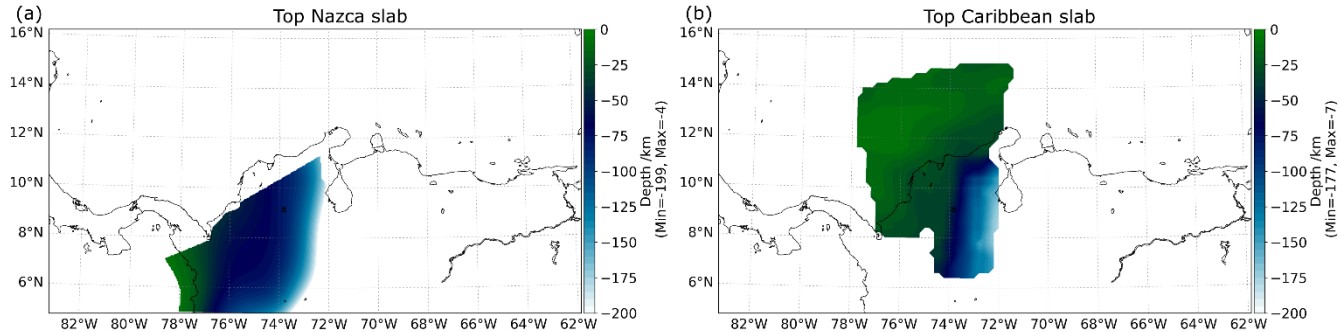


**Figure 5: Depth to the top of the slabs tested in the lithospheric-scale gravity models. (a) Nazca geometry from the Slab2 dataset (Hayes et al., 2018). Note that this geometry does not differentiate the Nazca (Coiba) and the Caribbean components in the slab. (b) Caribbean slab published by Mora et al. (2017).**

Table 1 summarises the densities assumed in the initial 3D lithospheric model for the different layers considered. This model

includes the gravity response of the 3D mantle densities in addition to the integration of the Nazca and Caribbean slabs geometries, as represented in Fig. 3 (f). When both slabs are integrated into the model, a density of 3163 kg m$^{-3}$ is obtained with the IGMAS+ inversion algorithm (Sæther, 1997; Schmidt et al., 2011).

### 3.3 Forward modelling of the gravity residuals

After the analysis of the mantle density effect on the modelled gravity was performed, the model which had the minimum

RMSE compared with the observed gravity anomalies of the EIGEN-6C4 dataset (Förste et al., 2014) was selected as the initial lithospheric configuration (Sect. 3.2).

The relative short wavelengths of the still remaining gravity residuals (observed gravity minus modelled gravity) associated to this model account for rather shallow density heterogeneities. Indeed, the initial model considered the oceanic and continental crusts as homogeneous layers with constant densities (see Table 1). Therefore, in order to have a better





representation of the crustal heterogeneities, the forward modelling of the gravity residuals of the initial model was performed in three main steps. First, the code "Harvester" of the Fatiando a Terra geophysical Python tool (Uieda et al., 2013) was used for defining the boundary between the upper and lower continental and oceanic crusts. Harvester plants "density seeds" at the base of the crust, and makes them grow vertically until the gravity residuals are minimised, defining the top of the lower crust. In our case, a density contrast of 300 kg/m$^3$ was assumed between the upper and lower crust for this inversion.

**Table 1. Summary of the datasets used to constrain the initial 3D lithospheric-scale model of the South Caribbean and north-western South American plate, and the density values (kg m$^{-3}$) assumed for each layer.**

| Layer | Density | Reference of structural layer |
|---|---|---|
| Water | 1040 | GEBCO (Weatherall et al., 2015) |
| Oceanic sediments | 2350 | CRUST1.0 (Laske et al., 2013) |
| Continental sediment | 2500 | |
| Oceanic crust | 2900 | Moho depth from GEMMA model |
| Continental crust | 2800 | (Reguzzoni and Sampietro, 2015) |
| Nazca slab | 3163 | Slab2 (Hayes et al., 2018) |
| Caribbean slab | | (Mora et al., 2017) |
| Mantle from Moho down to 50 km depth | 3200 | - |
| Mantle from 50 down to 200 km depth | 3D density solution | SL2013sv tomographic model (Schaeffer and Lebedev, 2013) |

Second, in the regions where the residuals (after the inversion with Harvester) were still considerably large (>30 mGal and <-30 mGal), bodies in the upper and lower crust within the oceanic and continental domains were defined (see Sect 4.1). The

delimitation of these bodies took into account the geologic evolution of the CLIP, and in general, the tectonics of the Caribbean provinces (for example well known volcanic arcs, previously reported underplated material, or extended proto-Caribbean crustal domains). Their densities were assigned depending on whether they matched a positive or negative residual, which suggest a mass deficit or excess in the initial model, respectively.

Finally, after implementing the previously mentioned two steps, large gravity residuals of medium size wavelengths were still

present in the oceanic domain of the Caribbean, which led us to define two high-density bodies in the uppermost oceanic lithospheric mantle (< 50 km) of the Colombian and the Venezuelan basins. At these depths, trade-offs between the crust and the shallow mantle may occur, affecting the reliability of the tomographic data.

## 4 Results

After testing the gravity response to different mantle configurations, the model which integrates all the regional scale

observations was selected as the initial lithospheric configuration (Table 1). The residuals associated to this model are depicted

in Fig. 6, which have a RMSE of 28.84 mGal, the minimum value of all tested models (supplementary material S1). The wavelength of these residuals indicates that they are mainly due to shallow heterogeneities that were not considered in the initial set up of the model. Therefore, the forward modelling of these residuals was used to derive density heterogeneities in a regional scale, present within the continental and oceanic crystalline crusts, as well as in the uppermost ($< 50$ km depth)

lithospheric mantle, as described in the next subsections.

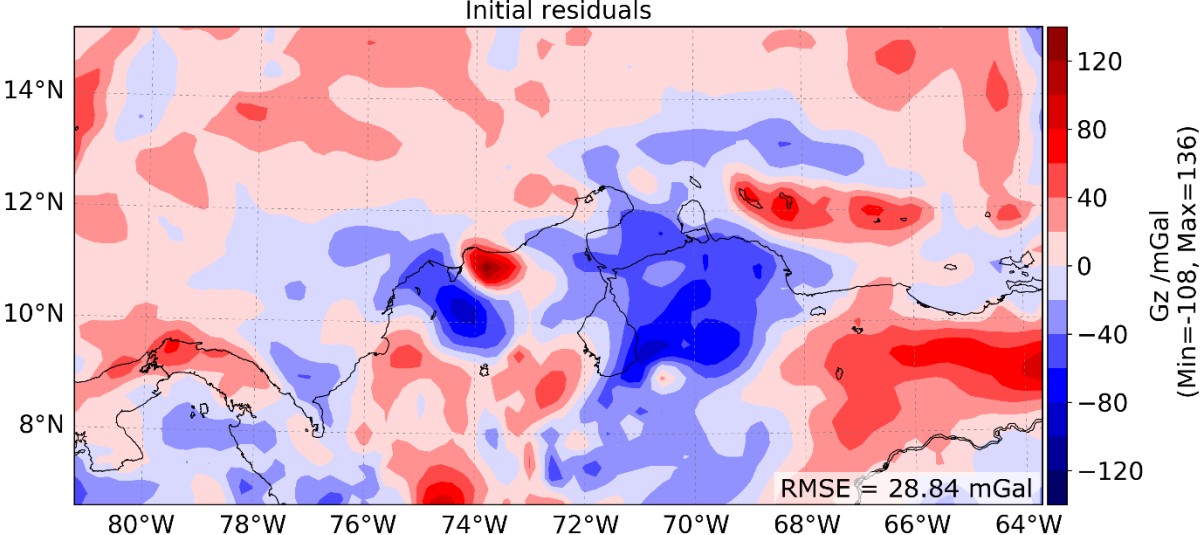

**Figure 6: Gravity residuals of the initial lithospheric configuration, assuming the densities in Table 1. Here, large misfits with a dominance of short (few km) and medium (hundreds of km) wavelengths suggest density heterogeneities in the shallower domains of the lithosphere, that were not considered in the initial set up of the model.**

**4.1 Crustal structure of the Caribbean and north-western South American plates**

Figure 7 (a) depicts the observed free-air gravity of EIGEN-6C4, at 10 km calculation height. This field has large positive values (>180 mGal) over the topographic highs of the Andes, as well as over the Santa Marta massif and the Panama microplate (Fig. 1). In contrast, negative gravity anomalies lie over the thick depocenters of the South Caribbean margin (Panama Deformed Belt, Magdalena Fan and South Caribbean Deformed Belt; Fig. 3 (b)), as well as over most of the continental basins

(sub. Lower Magdalena, Barinas-Apure and Maracaibo; Fig. 3 (c)).

As described in Sect. 3.3 the first two steps on the forward modelling process consisted on including crustal heterogeneities aiming to fit the gravity field. Fig. 7 (b) shows the residuals associated to a model in which the gravity signal of the water, sediments, heterogeneous crystalline crust and mantle (deeper than 50 km) were considered. Here, two positive anomalies remain in the Colombian and Venezuelan basins. Thus, the third step on the forward modelling required the addition of two

subcrustal bodies to compensate the gravity misfit in these domains. These bodies are located between the Moho and 50 km depth and have an average density of 3242 kgm$^{-3}$.





As can be seen in Fig. 7 (c), after forward modelling the gravity residuals including the subcrustal bodies, the modelled gravity response resembles the features previously described for the EIGEN-6C4 field. The new residuals have a RMSE of 17.45 mGal, and have been minimised for the entire study area (Fig. 7 (d) and S5 (f)) compared with the residuals of the initial model 300 (Fig. 6 and S5 (d)). With the new lithospheric configuration, a decrease of approximately 39.5% in the RMSE is reached.

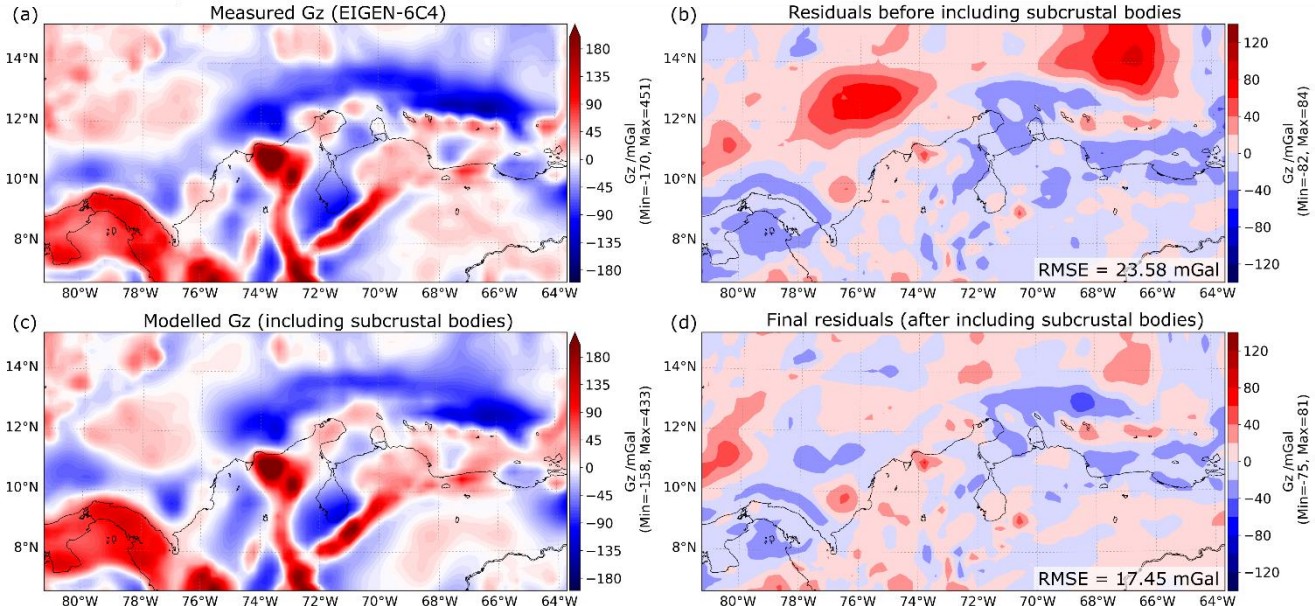

**Figure 7: Gravity anomalies (Gz): (a) measurements available in the EIGEN-6C4 dataset at 10 km height (Förste et al., 2014; Ince et al., 2019). (b) Residuals of a model in which the subcrustal bodies are not included. (c) Gz after forward modelling the gravity residuals of the initial lithospheric configuration. The modelling included the definition of crustal and subcrustal bodies (see text for 305 details). (d) Final residuals obtained with the lithospheric structure described in Table 2.**

This new lithospheric configuration aims to represent the complex tectonic setting of the Caribbean crust by including six different oceanic layers, with low- and high-density bodies (Fig. 8), as summarised in Table 2. The oceanic upper crust has been modelled with an average density of 3000 kg m$^{-3}$ and its maximum thickness reaches up to 9.55 km (Fig. 8 (a)). Moreover, low- and high-density bodies were defined within this layer. A low-density domain is present below the Aves Ridge (yellow 310 polygon), with an average density of 2900 kg m$^{-3}$, while a high-density body was found below the Venezuelan basin (magenta polygon), whose average density is 3300 kg m$^{-3}$.

The oceanic lower crust (Fig. 8 (b)) has been modelled with an average density of 3100 kg m$^{-3}$. Within this layer, widely distributed high-density bodies are required to fit the gravity field both for the Colombian and the Venezuelan basins (magenta polygons), with an average density of 3250 kg m$^{-3}$. The density assigned to the low-density bodies located in the Aves Ridge 315 and the South Nicaraguan Rise is 3000 kg m$^{-3}$. The maximum thickness of this layer is 24.7 km, towards the west of the Aves Ridge.

As a reference, Fig. 8 (a) includes the places where Mauffret and Leroy (1997) reported a thinner oceanic crust (< 10 km, black dashed lines). Similarly, Fig. 8 (b) depicts the areas where extended Proto-Caribbean crust has been reported in the Colombian



and the Venezuelan basins (black lines, numbers 1 and 2). Note the relation between these extended domains and the thin (<

1 km) or almost non-existent crust present in the structural model.

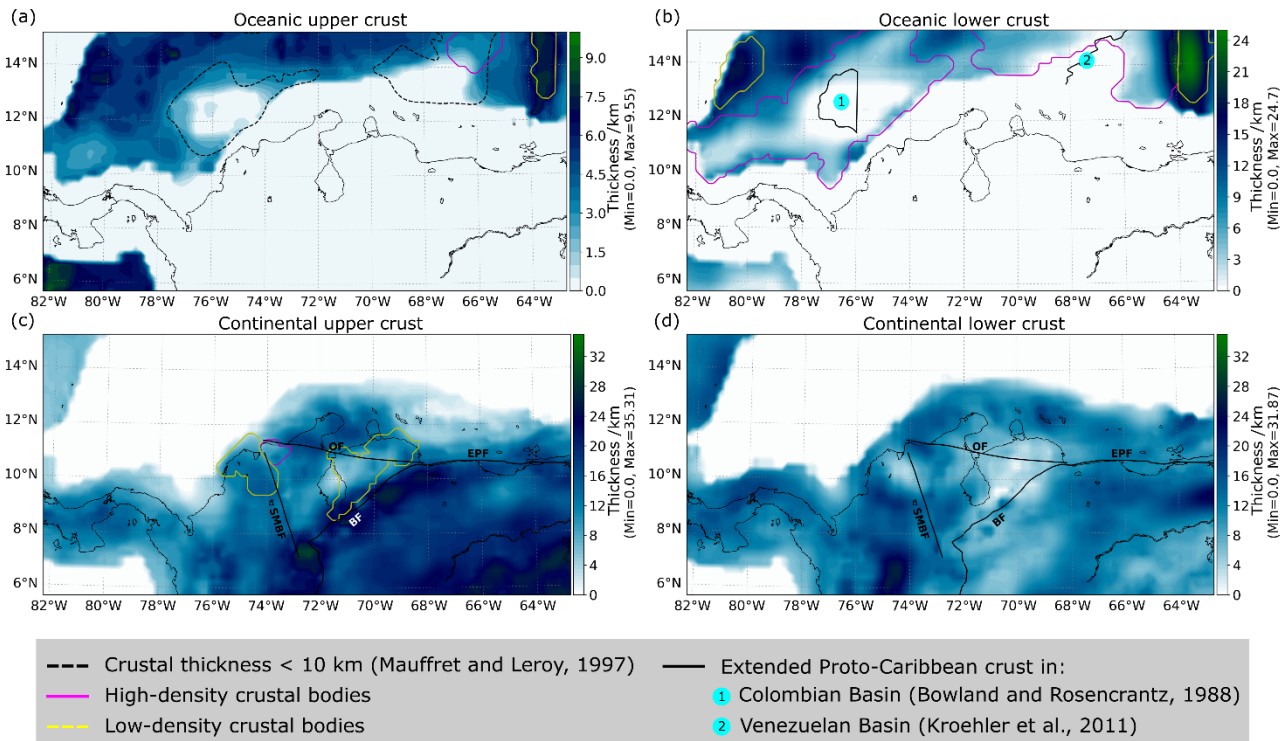

**Figure 8: Thickness of the (a) oceanic upper crust, (b) oceanic lower crust, (c) continental upper crust, and (d) continental lower crust. Dashed polygons in panel (a) represent places where the oceanic crust is thinner than 10 km (Mauffret and Leroy, 1997). Black polygons in panel (b) depict areas with smooth basement (B" horizon), associated with extended Proto-Caribbean crust in the**
**Colombian Basin (Bowland and Rosencrantz, 1988) (number 1), and in the Venezuelan Basin (Kroehler et al., 2011) (eastward of the black line, number 2). Main continental faults are shown as black thick lines in panels (c) and (d). The final lithospheric model includes different high- and low-density bodies in the oceanic and continental crusts, depicted as magenta and yellow polygons, respectively. Acronyms as in Fig. 1.**

The continental crust has been split into four different sublayers. The continental upper crust (Fig. 8 (c)) has the maximum
thickness (up to 35.3 km) and an average density of 2750 kg m$^{-3}$. Two low density bodies where defined within this layer in order to improve the gravity fit. These areas include parts of the Lower Magdalena, Maracaibo and Falcon basins (yellow polygons). Such bodies have an average density of 2600-2650 kg m$^{-3}$. Additionally, one high density body has been integrated into the structure of the upper continental crust (magenta polygon). It has been modelled with an average density of 3000 kg m$^{-3}$, and is located below the Santa Marta massif, bounded by the Oca and the Santa Marta-Bucaramanga faults.

Finally, the lower continental crust has been defined as a complete unit, with an average density of 3070 kg m$^{-3}$, and a maximum thickness of 31.8 km, reached in the southern part of the eastern cordillera.

Assuming this new lithospheric configuration, the average crustal density highlights high- and low-density domains that correspond with thin and thick crystalline crust, respectively (Fig. 9). In general, it is possible to recognise that lighter crust



(Fig. 9 (a)) is associated with the continental domains, as well as with portions of the oceanic arc-like Nicaraguan Rise (Lewis
et al., 2011), and the extinct arc of the Aves Ridge (Christeson et al., 2008). On the other hand, the highest densities are
concentrated in the Colombian and the Venezuelan basins. Two regions where the calculated thickness of the oceanic crust is
"virtually zero" are depicted with as white areas in Fig. 9 (a) and as magenta polygons in Fig. 9 (b), and will be discussed in
Sect 5.2.

Within the continental domain, two low-density regions ($< 2700$ kg m$^{-3}$) are inferred based on the model results: the Lower
Magdalena Valley basin, whose basement has been described as mainly felsic (Mora-Bohórquez et al., 2017); as well as parts
of the Falcon and Maracaibo foreland basins. These continental basins are characterised by a thin crust ranging between ~10
and 20 km (Fig. 9 (b)). In contrast, the crust of the Santa Marta massif has an average density of ~3000 kg m$^{-3}$, which is in
agreement with a cratonic origin with igneous intrusions, as described by Montes et al. (2019).

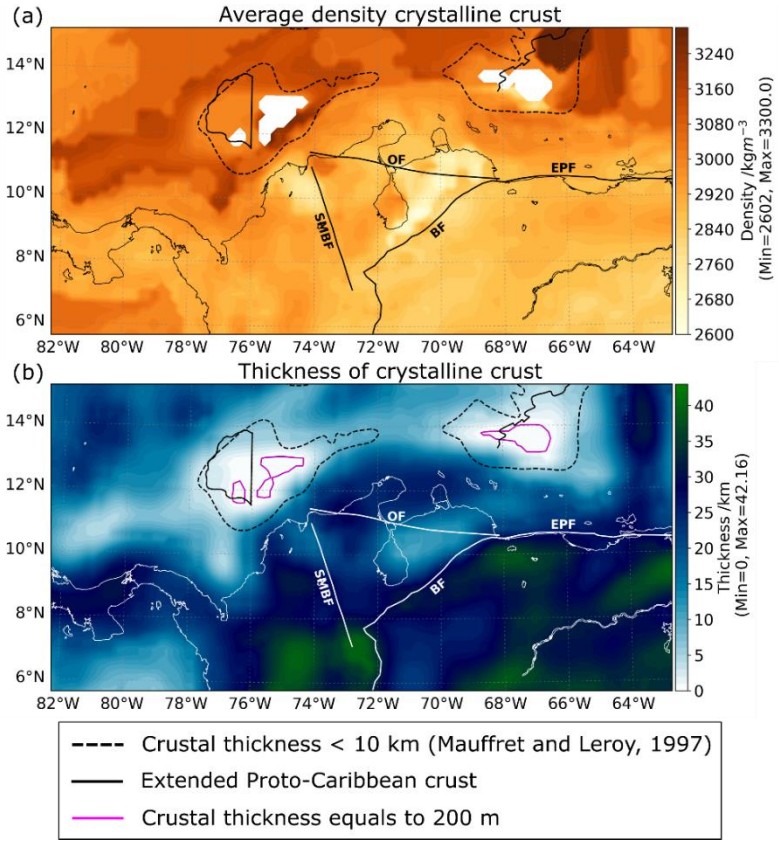

**Figure 9: The average density of the crystalline crust obtained after forward modelling the gravity residuals highlights areas where
the high-density material of the CLIP formation strongly affected the Proto-Caribbean crust. (a) Average density of the South
Caribbean and north-western South American crusts based on the forward modelling of the gravity residuals. The crustal domains
summarised in Table 2 have been included. White domains represent areas were the thickness of the oceanic crust is virtually zero,
according to the integrated datasets. (b) Thickness of the crystalline crust used in the data-integrative 3D model (see Sect. 3.1).**
**Magenta polygons are contours of 200 m thickness. Other polygons as depicted in Fig. 10. Acronyms as in Fig. 1.**



## 4.2 Oceanic upper mantle structures

In order to fit the gravity, two subcrustal bodies were additionally required and integrated into the oceanic upper mantle ($< 50$ km depth) (Fig. 10). According to the model results, these bodies might reach an average density of $\sim 3300$ kg m$^{-3}$, if

they are placed from the Moho down to 25 km depth (see Fig. S6). Alternatively, an average density of 3242 kg m$^{-3}$ is required if they extend down to 50 km depth. Due to the fact that a similar high-velocity trend - that would convert to comparable densities- can be followed in the tomographic data integrated into the lithospheric model, down to 75 km depth, the last scenario is selected hereafter as the final lithospheric configuration. These high-density bodies spatially correlate with the areas of thin crust according to Mauffret and Leroy (1997) (dashed lines in Fig.10), but also include the regions where original Proto-

Caribbean crust has been identified by Bowland and Rosencrantz (1988) in the Colombian Basin, and by Kroehler et al. (2011) in the Venezuelan Basin (black lines in Fig. 10 -number 1 and 2, respectively). Thus, the high-density bodies are obtained in areas where the already shallow crust-mantle boundary is insufficient to account for the excess mass needed to fit the observed gravity.

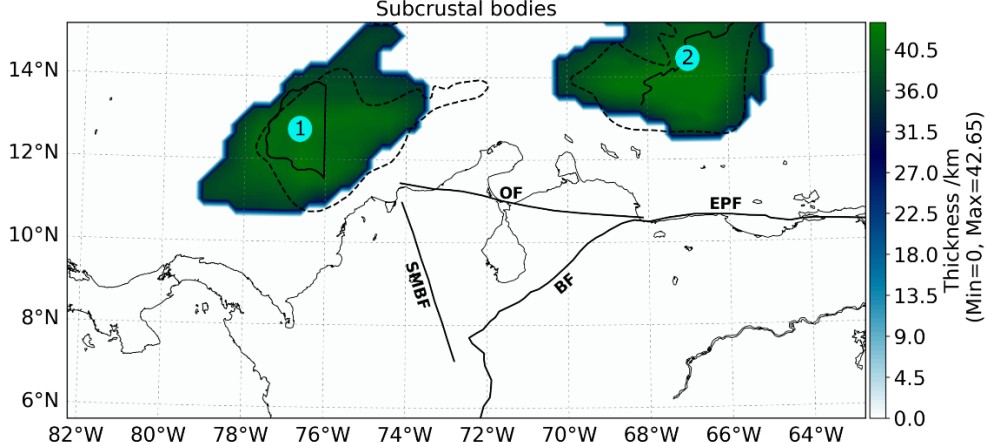

**Figure 10: Two high density subcrustal bodies are required to fit the gravity field. These bodies have a spatial correlation with the thin crust in the Colombian and the Venezuelan basins. Polygons as described in Fig. 8. Acronyms as in Fig. 1.**

A summary of the final lithospheric structural and density model is presented in Table 2. As mentioned in Sect. 3.3, the gravity inversion was focused on improving the structural resolution for the oceanic and continental crust, and the uppermost oceanic

mantle; therefore, the remaining lithospheric layers were preserved as in the initial (reference) model.

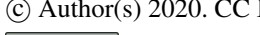

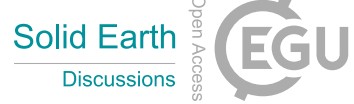

**Table 2. Summary of the layers and density values that integrate the final structural model of the South Caribbean and north-**
**western South American lithosphere.**

| Layer | Modelled density /kg m⁻³ |
|---|---|
| Water | 1040 |
| Oceanic sediments | 2350 |
| Continental sediments | 2500 |
| Oceanic upper crust | 3000 |
| Low-density upper crustal body (Aves Ridge) | 2900 |
| High-density upper crustal body (Venezuelan basin) | 3300 |
| Oceanic lower crust | 3100 |
| Low-density lower crustal bodies (Aves Ridge and South Nicaraguan Rise) | 3000 |
| High-density lower crustal bodies (Colombian and Venezuelan basins) | 3250 |
| Continental upper crust | 2750 |
| Low-density bodies upper continental crust (Eastern cordillera and Lower Magdalena basin) | 2600 - 2650 |
| High-density body upper continental crust (Santa Marta Massif) | 3000 |
| Continental lower crust | 3070 |
| High-density subcrustal bodies down to 50 km depth | 3242 |
| Mantle down to 50 km | 3200 |
| Nazca and Caribbean slabs | 3163 |
| Mantle from 50 km to 200 km | 3D density solution |

## 5 Discussion

The gravity response of a 3D lithospheric-scale model of the Caribbean and northern South American plates - including up-to-date geophysical information - provides additional constrains regarding density heterogeneities in the lithosphere. Although a heterogeneous crust was integrated into the model in order to reduce the misfits between modelled and observed gravity, two

large positive residuals in the oceanic domain (Fig. 7 (b)) suggest additional density heterogeneities in the oceanic upper mantle (< 50 km depth), which was initially modelled with a constant density. Our approach indicates that the results of the gravity inversion are beyond the non-uniqueness of gravity, because: 1. The (heterogeneous) crustal densities cannot be further





increased; 2. A sensitivity analysis to different mantle distributions was initially performed (supplementary material S1); and

3. A high-density trend is also evidenced in the tomographic data down to 75 km depth in the Caribbean region.

In the next subsections, the implications of the high-density upper mantle bodies are discussed, as possible plume conduits of

the CLIP currently preserved in the Caribbean upper mantle (Sect 5.1). Additionally, different kinematic reconstructions back

to 90 Ma were explored to evaluate the hypothesis that the CLIP formed above the Galápagos hotspot. Remaining implications

of the general crustal configuration are tackled in Sect 5.2.

## 5.1 High density mantle trend: preserved material of the CLIP plume?

In the Caribbean oceanic upper mantle, two high-velocity (and therefore high-density) domains can be followed, at least down

to 75 km depth, according to the S-wave tomographic model of Schaeffer and Lebedev (2013) (Fig. 4, S3 and S4), that has

been integrated in the initial (reference) density model of the Caribbean. After forward modelling the gravity residuals to

identify density heterogeneities in the crystalline crust, the residuals were still large enough to call for density variations on

the uppermost oceanic mantle (< 50 km depth), which was initially modelled with a constant density of 3200 kg m$^{-3}$ (Table 1).

Accordingly, the results suggest that two subcrustal bodies with positive density anomalies are present from the Moho down

to 50 km depth, which spatially correspond with a fast velocity mantle trend observed in the tomographic data (Fig.11).

Assuming the thickness and spatial distribution shown in Fig. 10, an average density of ~3242 kg m$^{-3}$ is required to fit the

gravity field. These bodies, approximately 440 km wide, show a strong spatial correlation with the thinner crust of Mauffret

and Leroy (1997) (black dashed polygons), and also with the stretched original Proto-Caribbean crust reported by Bowland

and Rosencrantz (1988) in the Colombian Basin (number 1), and by Kroehler et al. (2011) in the Venezuelan Basin (eastward

of the black line, number 2).

Nevertheless, it is necessary to point out that the density of these subcrustal domains might not necessarily be exactly

3242 kg m$^{-3}$. In fact, if their vertical extension is assumed from the Moho down to 25 km depth, the density required to fit the

gravity increases to 3300 kg m$^{-3}$, resulting in similar gravity residuals (see Fig. S6) as the ones shown in Fig. 7 (d). This

illustrates the range of non-uniqueness inherent to gravity modelling but also clearly demonstrates that the need for an excess

of mass in these domains is robust.

In both scenarios, the subcrustal bodies might have seismic velocities similar to that of a "normal" mantle, making their

interpretation as mantle plume-related bodies difficult, especially in the context of early seismic experiments carried out in the

past. For example, Mauffret and Leroy (1997) interpreted seismic velocities of 8.1 km s$^{-1}$ as normal mantle velocities, based

on isolated sonobuoy measurements both in the Colombian and the Venezuelan basins (see Figure 22 (a) in their publication).

The presence of shallow subcrustal bodies has been identified for different plateaus and regions where plume interaction with

old crust is inferred (see McNutt and Caress, 2007). Some examples include a 4 km thick and 200 km wide subcrustal body

found in the Hawaiian-Emperor seamount chain by Watts et al. (1985). According to their interpretation, this body has

unusually high seismic velocities for a "normal" lower oceanic crust, ranging between 7.4 and 7.8 km s$^{-1}$. More recently, Deng



et al. (2014) reconstructed a preserved, lithospheric-scale magmatic intrusion associated with the evolution of the Emeishan
Large Igneous Province down to 120 km depth.

In the Caribbean region, underplated material associated with picritic and ultramafic cumulates with seismic velocities of up
to 8 km s$^{-1}$ have been previously identified (e.g. Driscoll and Diebold, 1999; Mauffret and Leroy, 1997). However, the high-
density (thus high-velocity) material as represented in Fig. 11 was never mapped before with such details and so deep beneath

the Colombian and Venezuela basins. In the presence of a mantle plume, intrusion of high-density material on the overriding
plate is expected. When the plume ceases, the new lithospheric system starts to cool down, and therefore, high-density bodies
can 'freeze' (François et al., 2018) and become part of the mobile lithosphere that moves over the hot asthenospheric mantle,
as expected by the plate tectonics theory. Therefore, we interpret these subcrustal bodies as preserved magmatic plume material
responsible for the CLIP formation.

Figure 11 shows a 3D perspective view of the preserved plume material assuming an S-wave velocity of 4540 ms$^{-1}$ as the
boundary of the high-density domains. The velocity contours at depths of 25, 50 and 75 km were taken as main constraints,
and the form of the body in between was obtained by morphing of these contours following Brunet (2020) and Brunet and
Sills (2015). A total volume of ~12 x 10$^6$ km$^3$ is calculated for what is interpreted as the CLIP fossil plume conduits. Figure
S7 shows the vertical relation between the subcrustal bodies, if a density of 3242 kg m$^{-3}$ is assumed for them, and the high-

density trend at each depth of the tomographic model. The presence of this high-density trend down to 75 km depth suggest
that it travelled with the mobile Caribbean lithosphere. Accordingly, Blanco et al. (2017) reported the Lithosphere-
Asthenosphere Boundary in the Caribbean region around 75-80 km depth.

Recently, Civiero et al. (2019) used Rayleigh Taylor models to simulate the 3D mantle convection for Newtonian and non-
Newtonian rheologies. Their results predict the development of plumelets in different stages of evolution, creating a complex

structure sometimes difficult to image using seismic waves. Thus, the observed positive density anomalies in the Caribbean
lithospheric mantle might be related with the fossil plume conduits of these type of Rayleigh Taylor-style instabilities, similar
to the ones nowadays observed in the Northern East African Rift by Civiero et al. (2019).



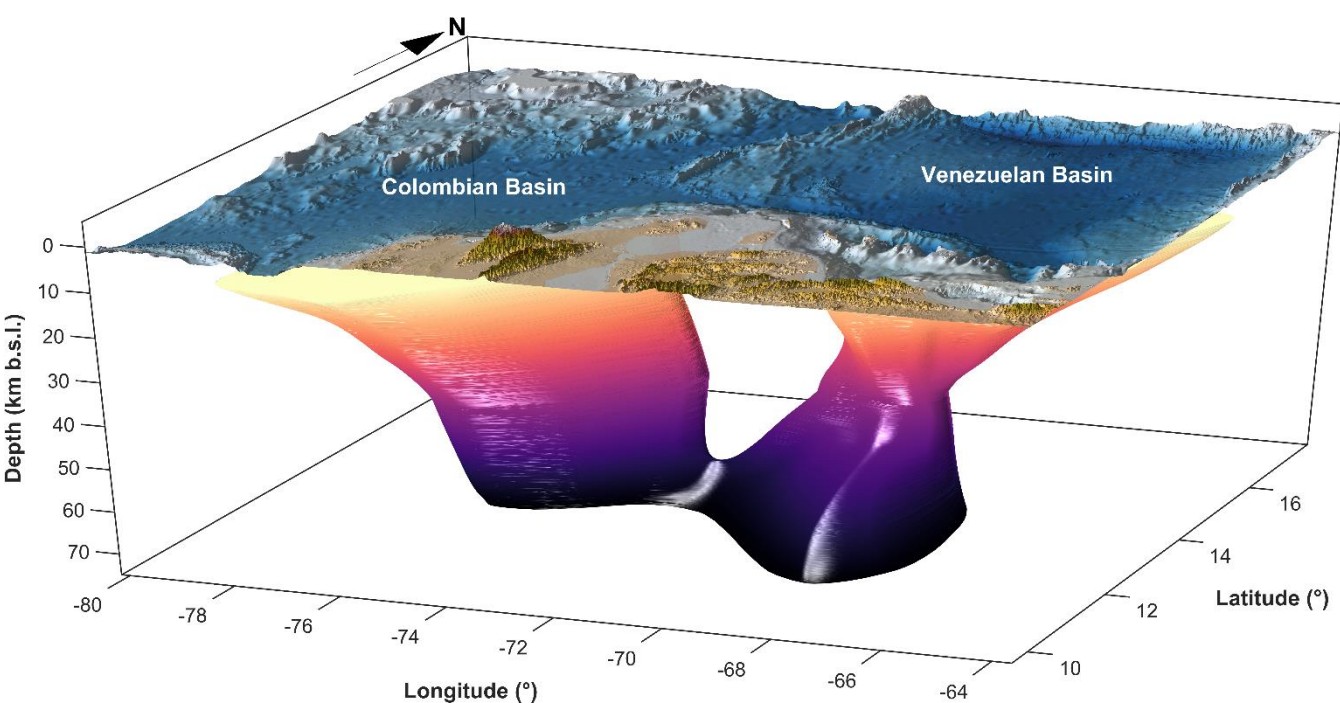

**Figure 11: 3D visualisation of the interpreted preserved plume conduits, assuming an S-wave velocity of 4540 ms$^{-1}$ as the boundary of the high-density material.**

## 5.2 Kinematic reconstructions of the Caribbean plateau and origin of the CLIP

Previous studies based on geochemical analyses of CLIP-related magmatic rocks have proposed the present-day Galápagos hotspot as the origin of the thermal anomaly responsible for the development of the CLIP (e.g. Duncan and Hargraves, 1984; Geldmacher et al., 2003; Kerr and Tarney, 2005; Pindell and Barrett, 1990; Thompson et al., 2004). From a kinematic perspective, Nerlich et al. (2014) obtained a good fit between the paleo-position of the Caribbean plate and the present-day position of the Galápagos hotspot at the time of plateau formation (c. 90 Ma). In their approach, the Caribbean plateau is assumed to have detached from the Farallon plate and be fixed relative to South America at 54.5 Ma ('docking' age), while using the finite rotation poles from Doubrovine et al. (2012) (global moving reference model) for the motion of the Pacific and African plates. However, short after, Boschman et al. (2014) published an updated kinematic reconstruction of the Caribbean region based on an extensive compilation of geological data from the entire region, which show a very different motion path of the Caribbean plateau relative to South America (cyan line in Fig. 12) than the model of Nerlich et al. (2014) (green line). A clear offset of the plateau to the east of the modern Galápagos hotspot at the time of CLIP formation (90 Ma) is observed in the model of Boschman et al. (2014), as previously reported by other authors (e.g. Pindell et al., 2006). More recently, Montes et al. (2019) reconstructed the tectonic evolution of the northern Andes-Caribbean margin and showed similar results as Boschman et al. (2014) (magenta line in Fig. 12), although a considerable difference in motion and rotation of the CLIP





between 90-60 Ma is evident. This difference is due to the fact that Montes' reconstructions are based on tectonic units in Panama and north west Andes terrains, while the Caribbean plate moves, bends and deforms internally to fit those inland proxies.

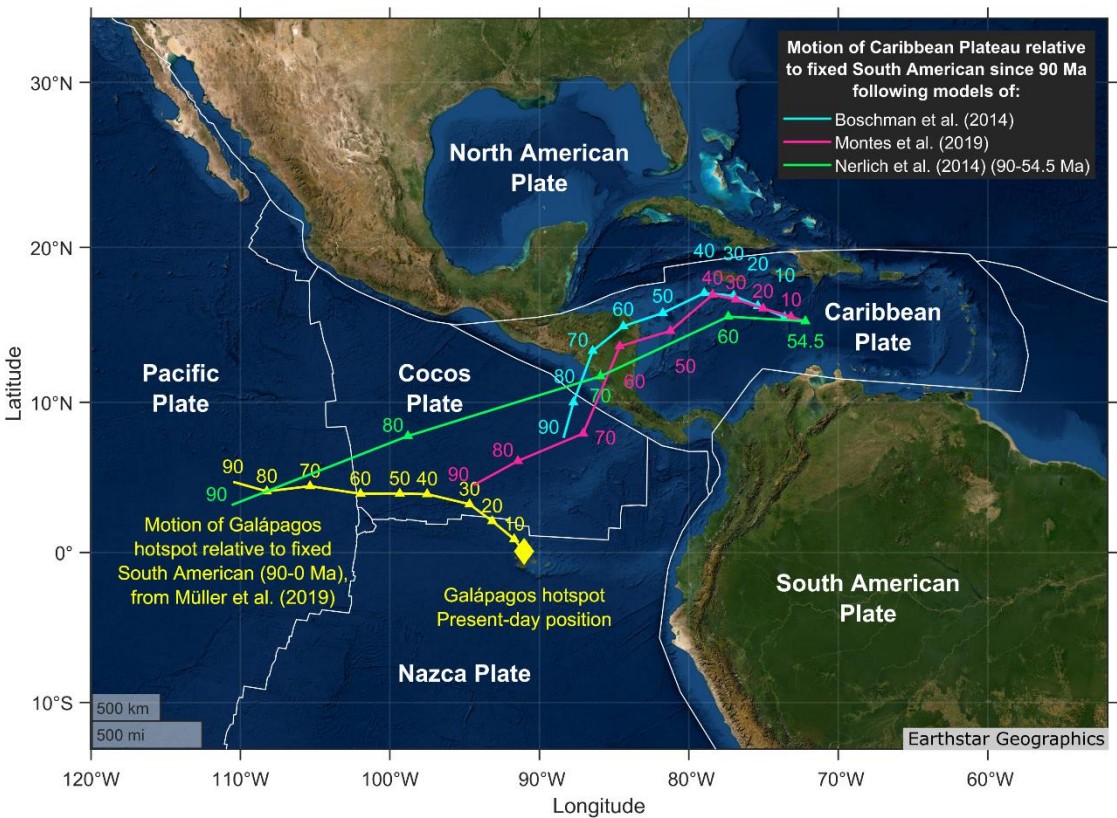


**Figure 12: Motion paths of Caribbean plate relative to South American plate (SAM) for three recently proposed reconstructions. Motion of present-day Galápagos hotspot relative to a fixed SAM is shown in yellow. A fit with Galápagos hotspot at time of plateau formation (90 Ma) is only obtained by Nerlich et al. (2014), assuming the Caribbean plate moving with Farallon plate from 90-54.5 Ma and fixed to SAM since 54.5 Ma. Regional reconstructions based on geological and geophysical datasets from the entire**
**Caribbean region (Boschman et al., 2014) and NW Andes (Montes et al., 2019a, 2019b) show, however, very different motion paths for the Caribbean plate relative to SAM and a clear offset with the Galápagos hotspot at 90 Ma.**

Most importantly, the absolute motion of the lithospheric plates relative to the Earth's deep interior is crucial to assess whether the CLIP formed above the Galápagos hotspot 90 Ma ago or not. Several absolute reference models have been proposed over the last decades, e.g. the moving Indian-Atlantic hotspots model (O'Neill et al., 2005), the fixed Pacific hotspots model (e.g.
Wessel and Kroenke, 2008), the true polar wander-corrected paleo-magnetic model (Steinberger and Torsvik, 2008), the subduction reference frame linking surface plate motions to subducted slab remnants mapped from seismic tomography (Van Der Meer et al., 2010), and the global hotspot moving model (Doubrovine et al., 2012). The geodynamic consistency of those models has been however recently questioned as they yield unreasonable velocities of slab advance and retreat, and of global net rotation of the lithospheric plates (Müller et al., 2016; Schellart et al., 2008; Williams et al., 2015). Tetley et al. (2019)
developed a data-optimised global absolute reference frame back to Triassic time that includes global hotspot track



observations and estimates of net lithospheric rotation and paleo-trench migration. This geodynamically consistent absolute plate model is implemented in the latest global plate motion model of Müller et al. (2019), which also includes the detailed regional reconstructions of Boschman et al. (2014) for the Caribbean region, and thus corresponds to our 'selected model' in the discussion below.

In this section, we aim to re-evaluate the hypothesis that the CLIP (and interpreted preserved plume conduits) formed above the Galápagos hotspot by testing six different plate motion configurations using the GPlates software (Müller et al., 2018). We used the recently published regional plate kinematic models of Boschman et al. (2014) and Montes et al. (2019a), for which the rotation files are available, and three different absolute reference frames, i.e. the "subduction reference model" of Van Der Meer et al. (2010), the "Global Moving Hotspot model" of Doubrovine et al. (2012), and the most recent "optimised global

reference model" of Müller et al. (2019), as summarised in Table 3. The rotation files of all tested models are available as supplementary material in Gómez-García et al. (2020) (see Data availability).

Figure 13 (a) presents the results of the different kinematic reconstructions back to 90 Ma, using the polygons of the subcrustal bodies in the Colombian and the Venezuelan basins (number 1 and 2, respectively). Here, white circles represent the plausible diameter of the plume head: 2000 km (Campbell, 2005) and 2500 km (Mauffret et al., 2001). Additionally, Fig. 13 (b) depicts

the motion paths of the centre of the Caribbean plateau since 90 Ma in absolute reference frame. In both figures, the present location of the Galápagos hotspot is represented by a yellow diamond, at 0 °N, 91 °W (Nolet et al., 2019). The main outcome of these reconstructions is the clear offset between the CLIP and the location of the hotspot at time of plateau formation (90 Ma) for all models, as previously reported by Boschman et al. (2014). In our 'selected model' (Müller et al., 2019), the CLIP centre was located c. 2500 km to the east of the present-day Galápagos hotspot 90 Ma ago.


**Table 3. Summary of the kinematic models used to test the origin of the CLIP in the Galápagos hotspot (0 °N, 91 °W -Nolet et al., 2019), and the corresponding distance to the centre of the Caribbean plateau at 90 Ma.**

| Regional kinematic model | Absolute reference frame | | |
|---|---|---|---|
| | Van Der Meer et al. (2010) | Doubrovine et al. (2012) | Müller et al. (2019) |
| | Distance from CLIP centre at 90 Ma to present-day Galápagos hotspot /km | | |
| Boschman et al. (2014) | 1946 | 3085 | **2505** |
| Montes et al. (2019a) | 1264 | 2352 | 1842 |

To evaluate the reasons behind this misfit, three possible sources of error should be discussed:

(1) The exact paleo-location of the Caribbean plate relative to South America is still not well known, especially prior to ~67 Ma (age of development of arc magmatism in Panama -Montes et al., 2012), as shown by the differences between the regional reconstructions of Boschman et al. (2014) and Montes et al. (2019a). However, this difference between the regional



reconstructions (c. 750 km) is less significant than the offset between the CLIP at 90 Ma and the present-day Galápagos hotspot, which ranges from 1264 to 3085 km, depending on the model used (c. 2500 km for our 'selected model', Table 3).

(2) The absolute motion of the tectonic plates can significantly change from one model to another. The smallest offsets between the centre of the CLIP at 90 Ma and the present-day Galápagos hotspot (1264 and 1946 km, Table 3) are obtained with the "subduction reference model" of Van Der Meer et al. (2010). The largest offsets (2352-3085 km), on the other hand, are obtained with the "Global Moving Hotspot model" of Doubrovine et al. (2012). We note that previous authors have obtained a fit between the modern Galápagos hotspot and the CLIP at 90 Ma using a fixed Pacific hotspots reference frame (Pindell and

Kennan, 2009) and/or a 'docking' of the Caribbean plate at 54.5 Ma (Nerlich et al., 2014). However, those models would require a larger differential motion of the Caribbean plate relative to South America than the motion described by Boschman et al. (2014) and Montes et al. (2019a). Moreover, as discussed above, those absolute reference frames are geodynamically inconsistent (e.g. Müller et al., 2016). Therefore we would favour the "optimised global reference frame" of Müller et al. (2019) that yields an offset of 1842-2505 km between the paleo-CLIP and the Galápagos hotspot at 90 Ma, if the latter is

considered fixed.

(3) The (potential) migration of the Galápagos hotspot since 90 Ma is not considered in the tested absolute reference frames. However, it is difficult to reconstruct a motion for this hotspot as the track observed today associated with the Cocos and the Carnegie ridges is younger than 20 Ma (Werner et al., 2003). Nevertheless, strong geochemical evidence suggests a correlation between the CLIP and the lavas of the present-day Galápagos hotspot. Particularly, the Hf-Nd isotopic composition is

interpreted as the Caribbean plateau representing the initial plume head of the Galápagos hotspot (i.e. Geldmacher et al., 2003; Thompson et al., 2004).

Steinberger and O'Connell (2000) proposed that the Easter Island hotspot migrated west- to south-westward at about 1-2 cm yr$^{-1}$ in a mantle reference frame, due to return mantle flow from the Andean subduction zone. Thus, if we assume a similar migration velocity for the Galápagos hotspot since the establishment of the western Caribbean subduction beneath northern

South America, at about 67 Ma (age of arc magmatism in Panama, Montes et al., 2012), its paleo-position would be c. 670-1340 km more to the E-NE back in Cretaceous time. If we subtract this motion from the offset of our 'selected model' (2505 km, using Boschman et al., 2014 and Müller et al., 2019, Table 4), a residual offset of c. 1165-1835 km between the migrated hotspot and the CLIP remains. A fit between the CLIP and the paleo-Galápagos would be obtained if we consider the possible large head of the plume during the Large Igneous Province formation (e.g. 2000-2500 km in diameter -Campbell, 2005;

Mauffret et al., 2001) as represented by the white circles in Fig. 13 (a).

To summarise, based on the argumentation above, the major offset between the paleo-position of the CLIP at 90 Ma and the present-day Galápagos hotspot can be interpreted either as: (1) The CLIP originated above the paleo-Galápagos hotspot only if this hotspot migrated significantly westward (> 1000 km at about 1-2 cm yr$^{-1}$ in a mantle reference frame) since the establishment of the western Caribbean subduction, and had a large plume head during the CLIP formation (~2000 km

diameter), or (2) The CLIP was formed by a different plume, which – if considered fixed - would be nowadays located below the South American continent (Fig. 13).



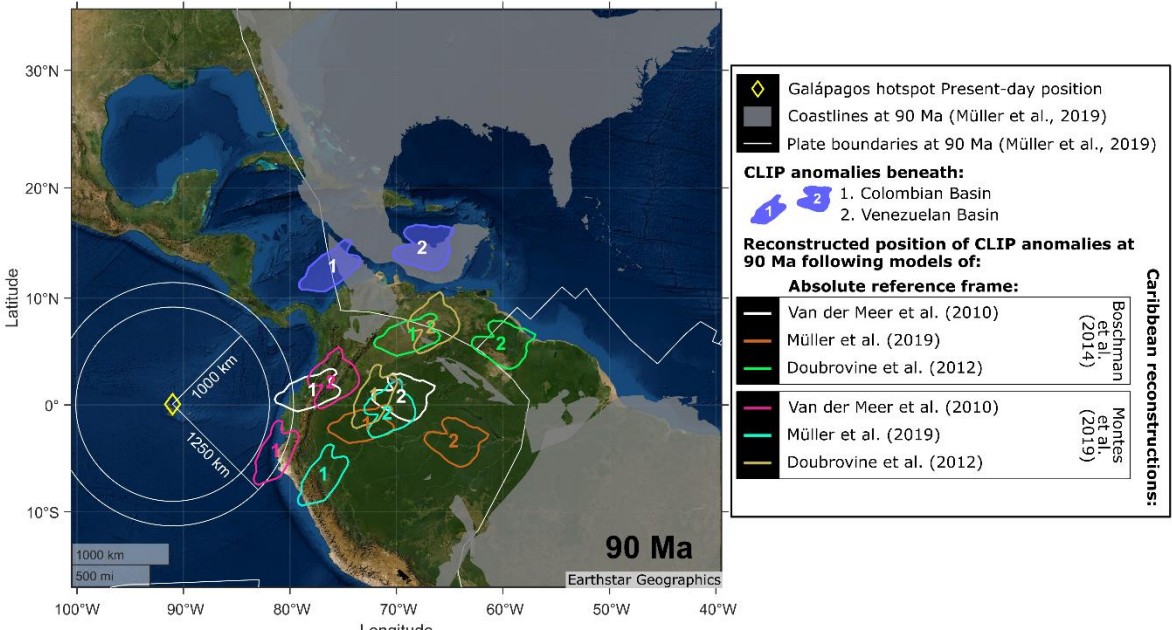

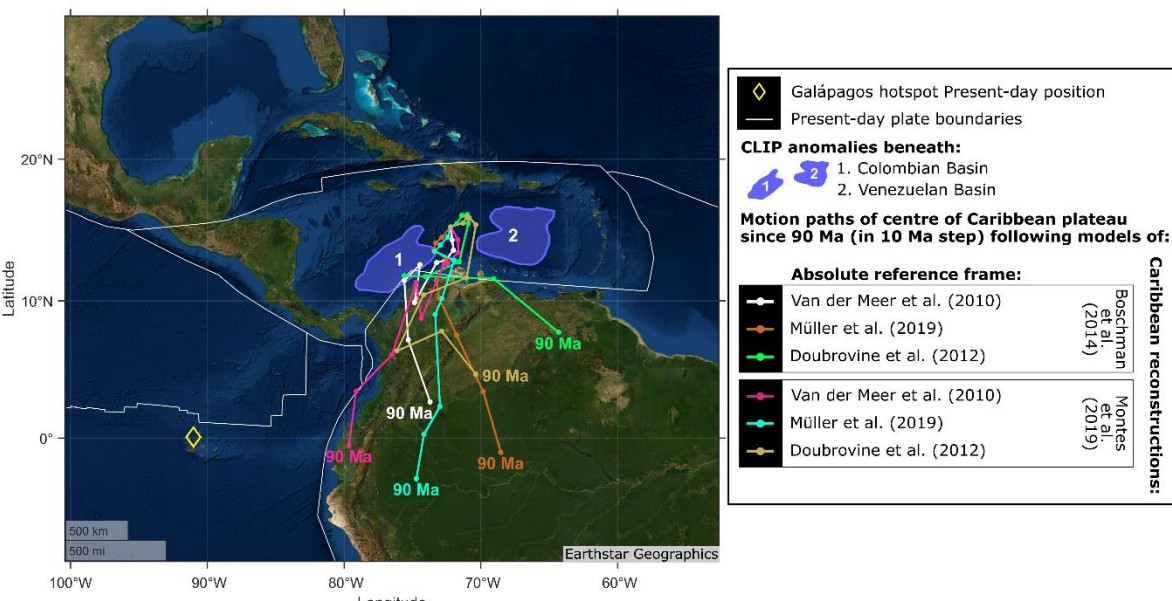

**Figure 13: Kinematic reconstructions using three different absolute reference frames and two regional Caribbean reconstructions show an offset between the positive mantle density anomalies and the location of the Galápagos hotspot back to 90 Ma. (a) Reconstructions at 90 Ma using the plume conduits polygons beneath the Colombian and the Venezuelan basins (numbers 1 and 2, respectively). White circles represent the plausible head of the Galápagos plume assuming a 2000 km and 2500 km diameter above a fixed plume. Present-day topography in the background for reference. (b) Motion paths of the centre of the CLIP since 90 Ma (in 10 Ma step). Different coloured lines depict the six different tested configurations.**



### 5.3 Implications for the tectonic development of the present-day Caribbean crust

The complex geologic development of the Caribbean crust involved volcanic mounds, sills, basalt flows, extensional episodes, and underplating of plume material (Driscoll and Diebold, 1999; Mauffret and Leroy, 1997), with heterogeneous, high MgO rocks (up to 28 wt% MgO, Kerr, 2014). The accreted portions of the CLIP allowed the reconstruction of the geochemistry and temperature conditions that the mantle plume had at the time of the interaction with the Proto-Caribbean crust (Arndt et al., 1997; Hastie and Kerr, 2010; Kerr, 2005; Van Der Lelij et al., 2010; Neill et al., 2011). Indeed, the high MgO rocks account

for magmas that migrated relatively fast from the mantle source (Kerr et al., 1998).

Geodynamic models support the evidence found in other Large Igneous plateaus related with a rapid, kilometre-scale uplift above the plume head. In these regions, the maximum uplift is also associated with lithospheric-scale extension in response to the buoyancy of the plume (e.g. Baes et al., 2016; Deng et al., 2014; François et al., 2018; He et al., 2003) and in most of the cases, the resulting crust is thicker-than-normal due to the magmatism above the plume head (Ernst, 2014). However, in the

CLIP history, the location of the plume head and the associated uplift event(s) have been poorly constrained. Nevertheless, Diebold and Driscoll (1999) recognised that the extension and magmatism in the Caribbean were synchronous processes.

As seismic profiles of sufficient quality and spatial coverage are not freely available, 3D gravity models were used to assess the average crustal density configuration on a regional scale (Fig. 9). The average crustal densities highlight high-density areas affected by the plume interaction with the Proto-Caribbean crust, which significantly contributed to the thicker than normal

crust that characterises this plate (Diebold and Driscoll, 1999; Driscoll and Diebold, 1999; Edgar et al., 1971; Ewing et al., 1960; Mauffret and Leroy, 1997). This is in agreement with the early tectonic model of the Caribbean (Diebold and Driscoll, 1999; Driscoll and Diebold, 1999).

However, it is important to note the spatial correlation between the thin crust (< 10 km) reported by Mauffret and Leroy (1997) and the extended Proto-Caribbean crust both in the Colombian (Bowland and Rosencrantz, 1988) and the Venezuelan basins

(Kroehler et al., 2011), with the subcrustal high-density bodies (Fig. 10). In the conventional tectonic model of the Caribbean of Diebold and Driscoll (1999), the CLIP plume has been considered to be located below the thicker than normal crustal domains (see Fig. 11 in Driscoll and Diebold, 1999), whereas the relationship between the extended domains of the Colombian and the Venezuelan basins with the CLIP evolution has not been well understood.

Geodynamic models show that the topographic uplift and subsidence patterns associated with the interaction of a mantle plume

with the oceanic lithosphere could be of several kilometres, depending on factors such as the plume size, mantle and lithosphere rheology and age of the overriding lithosphere (Burov and Cloetingh, 2009; François et al., 2018). Similarly, several studies suggest geological evidence of a rapid (~3 Ma) crustal doming prior to the eruption of basalts (e.g.: Deng et al., 2014; He et al., 2003; Li et al., 2014; Zhang et al., 2016).

The previously described arguments led us to propose that the anomalously thin crust above the plume conduits in the

Colombian and the Venezuelan basins correspond to the centres of uplift. However, the lack of evidence that supports underplated material beneath these basins needs to be explained, and we hypothesise two options: (1) The magmatism occurred





everywhere, but the uplifted part got eroded, as some geological evidence suggest for CLIP-related subaerial volcanism (Buchs et al., 2018). Indeed, the erosion of the Large Igneous Plateaus (LIP) has been widely recognised in other plateaus worldwide, including extreme cases where the preserved structure includes areas of no basalts in the centre of the plume head (Ernst et al.,

2005). (2) There was an offset between the plume head and the location of the basalt flows, and probably the magma paths were preferentially located along pre-existing lithospheric weakness zones (Buchan and Ernst, 2018; Ernst et al., 2019).

Based on the spatial correlation between the interpreted plume conduits and the extended crustal domains in the Colombian and the Venezuelan basins, we propose a modification to the tectonic model of the Caribbean, as described in Fig. 14. This cartoon is based on the tectonic model proposed by Driscoll and Diebold (1999), and on the present configuration of the

Venezuelan Basin (Fig. 2, 13 and 14 in Diebold and Driscoll (1999)).

The initial configuration of the Proto-Caribbean crust consisted of a "normal" Farallon plate located in the east Pacific (Fig. 14 (a)). In Early Cretaceous time, the plate started to interact with a mantle plume, creating rapid uplift and extension above the plume head (Fig. 14 (b)). In the centre of uplift, extensional structures such as horst and grabens may have formed. Subsequently, the early stage basalts (c. 90 Ma) spilled over the modified crust, preferentially flowing through predefined

weakened regions accompanied by lateral and vertical redistribution of magmas (Ernst et al., 2019). In this stage, underplating of ultramafic cumulates started to contribute to the increase in crustal thickness (Fig. 14 (c)). The lack of evidence for volcanism intruding the Proto-Caribbean domains of the Colombian and the Venezuelan basins today remains enigmatic (see question mark in Fig. 14 (c)). The thinning of the lithosphere above the plume head should be associated with decompression melting and magmatism, which to our knowledge, have never been reported in those areas. Therefore, either the seismic

reflection experiments have not been successful in mapping these structures, or the basaltic flows were later eroded (Ernst, 2007, 2014), or the episodes of basalt flows took place only in the surrounding areas (with an offset of the plume head) along inherited weak portions of the lithosphere, as Ernst et al. (2019) propose in their plumbing system model. Finally, a late stage (c. 76 Ma) of magmatism may have reactivated the former plume conduits and/or formed new ones. The frozen plume conduits migrated together with the Caribbean plate from the Pacific, forming the present lithospheric configuration of the Caribbean

plate, including the rough/smooth basement morphology described by Driscoll and Diebold (1999) (Fig 14 (e)). Parts of the CLIP were accreted along the margins, thus, just the preserved lithosphere is nowadays forming the Caribbean Sea (Fig. 14 (e)).

In conclusion, the preserved extended Proto-Caribbean crust in the Colombian and the Venezuelan basins might account for the centres of (kilometre-scale) uplift. The plume-related high-density bodies present in the uppermost 75 km of the Caribbean

lithosphere may have contributed significantly to thermal subsidence of these basins, as suggested by the large water depths (up to 5 km) in the Venezuelan Basin (see Fig. 3 (a) and Fig. 10, around 14° N and 66-68° W). Moreover, they might behave as rigid domains nowadays, affecting the overall behaviour of the Caribbean plate in response to regional tectonic stresses.





(a) Proto-Caribbean lithosphere in Early Cretaceous

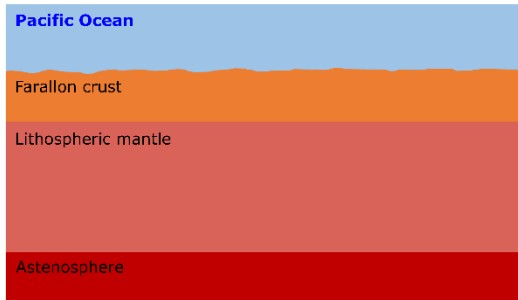

(b) Rapid uplift and extension prior to basalt flows

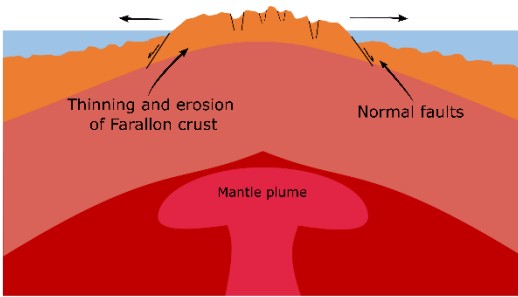

(c) ~91–88 Ma basalt flows and possible erosion

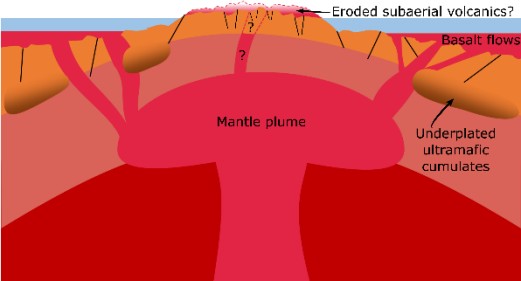

(d) ~76 Ma basalt flows, additional extension and subsidence

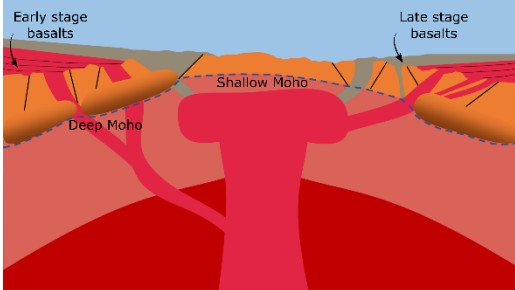

(e) Configuration of the Caribbean lithosphere

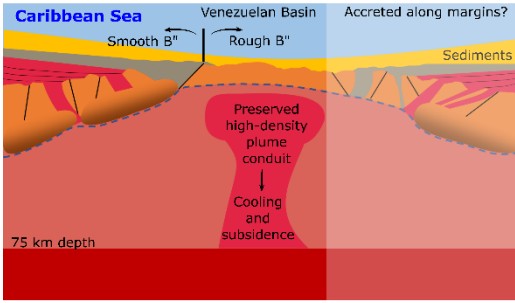



**Figure 14: Cartoon illustrating the tectonic development of the Caribbean crust (Venezuelan Basin) due to the interaction of the Farallon plate with a mantle plume. (a) Proto-Caribbean lithosphere with a "normal" crustal thickness in Early Cretaceous time. (b) Rapid uplift and thinning of the lithosphere due to the income of hot, buoyant plume material. This stage also should have included the erosion of the uppermost crustal structures (e.g. Ernst, 2007, 2014) such as horsts and grabens associated with the stretching of the crust. Thus, only the main faults are preserved. (c) The early stage basalts are spilled over the modified Proto-Caribbean crust, probably including subaerial erosion of the ones spilled over the emerged portion of the crust. In this stage, the underplated ultramafic cumulates contributed to crustal thickening. Question mark indicates that the existence of magma conduits beneath the proposed centre(s) of uplift remains unclear. (d) The late basalts covered the crust, which experienced additional extension and subsidence. In this stage, a differential Moho depth becomes more evident. (e) In the present-day configuration of the Caribbean plate, the smooth basement (B" horizon) is mainly preserved in the areas outside the uplift centre. The deepest Venezuelan and Colombian basins correlate with the extended-most crustal domains, where the doming was more pronounced. Due to the north – north-eastward migration of the Proto-Caribbean plate, fragments of the CLIP were accreted to the continental margins of the South American plate.**

## 6 Conclusions

3D lithospheric-scale, data-integrative models of the Caribbean and north-western South American plates reveal density heterogeneities both in the crystalline crust and the upper mantle, which are associated with the development of the CLIP. The fitting of the measured gravity anomalies requires the presence of two relatively high-density (higher than the surroundings) subcrustal bodies, ranging between 3242 and 3300 kg m$^{-3}$ (depending on their a priori defined thickness). These high-density domains can be followed down to 75 km depth, over approximately 400 km wide area, and are interpreted as the preserved, remaining material of the plume conduits responsible for the CLIP formation. Such fossil plume conduit had never been identified before.

The observed two branches of high-density mantle might be related with the magma conduits of the two main pulses of magmatic activity, broadly recognised in the tectonic model of the CLIP formation, at ~91–88 Ma and ~76 Ma. Assuming a S-wave velocity of 4540 m s$^{-1}$ as the boundary of the high-density mantle material, a total volume of ~12×10$^6$ km$^3$ is calculated for the Caribbean fossil plume. This volume is ~9 orders of magnitude higher than excess magma volume reported by Kerr (2014), for the 'in situ' plume material that currently forms the Caribbean crust. Nevertheless, future Ocean Bottom Seismometers (OBS) campaigns would be required to obtain a more detailed image of the Caribbean upper mantle, providing new insights on the evolution of these intriguing, vast magmatic processes.

Surprisingly, the high-density anomalies are not found beneath the thick CLIP crust and related basaltic flows, but below the thinned Proto-Caribbean crust of the Colombian and the Venezuelan basins. We propose that these two areas underwent significant extension during uplift above the head of a rising mantle plume and that magmatism either took place mostly on the side of the plume head, possibly along pre-existing weakness zones in the lithosphere; or magmatism was also present over these thinned domains, but eroded probably in a subaerial environment.

Furthermore, the hypothesis that the CLIP formed above the Galápagos plume was revisited. Kinematic reconstructions in absolute reference frame show offsets in the range of ~1200 to 3000 km (depending on the model used) between the location of the CLIP at the time of plateau formation (90 Ma) and the present-day Galápagos hotspot. These misfits suggest that either: (1) The CLIP originated above the paleo-Galápagos hotspot only if it migrated significantly westward (> 1000 km at about 1-

2 cm yr$^{-1}$ in mantle reference frame) since the onset of the western Caribbean subduction, and had a large plume head during the CLIP formation (~2000 km diameter), or (2) The CLIP was formed by a different plume, which – if considered fixed - 
would be nowadays located below the South American continent. Nevertheless, the existing strong geochemical evidence which supports the connection between the CLIP and the Galápagos hotspot favour the former and may highlight remaining pieces of the Caribbean "puzzle" that are not being considered in the kinematic models nowadays available.

Therefore, future efforts should focus on the improvement of the (1) imaging with more detail the structure of the Caribbean lithosphere, (2) plate-tectonic reconstructions of the Caribbean region and their relation with the probably not fixed Galápagos 
hotspot, and (3) testing plume-lithosphere interactions (cooling and motion of plume conduits with the lithosphere) with geodynamic models.

Finally, the workflow presented in this manuscript can be implemented in other plateaus worldwide as it relies on freely available, high-resolution geophysical information.

**Data availability**

The data to reproduce the results using the final model (M4) is available in Gómez-García et al. (2020) https://dataservices.gfz-potsdam.de/panmetaworks/review/9de5cf08ed825b2ee66e7418a16795afabb56bc3fa3dd5163321d6c0eed9ba91/. This data repository also includes the rotation files of the different kinematic reconstructions mentioned in this manuscript.

**Author contribution**

AMGG developed the initial research idea, carried out the collection, processing and integration of the data for the lithospheric structural and density models, plotted most of the figures, and wrote the initial draft of the manuscript. ELB carried out the kinematic reconstructions and most of the related figures. She also contributed in the literature review and in the discussion of the manuscript. MS and GM critically contributed with ideas during the development of the project, following closely the workflow steps and discussing the results. DA took part in the initial development of the structural models and contributed to 
the 3D visualisation of the positive density anomalies as well as the figures related to the kinematic reconstructions. All the authors actively contributed to the revision and improvement of the draft version of the manuscript.

**Competing interests**

The authors declare that they have no conflict of interest.



**Acknowledgements**

We would like to thank Prof. Camilo Montes (Universidad del Norte) for the fruitful discussion about the GPlates reconstruction in the Caribbean. AMGG is thankful with Dr. Álvaro González (CRM, Barcelona) for the discussion regarding the tectonic development of the Caribbean. AMGG was partially supported by COLCIENCIAS, Fundación para la Promoción de la Investigación y la Tecnología, CEMARIN, and DAAD research grants.

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
