# Peer review of "The preserved plume of the Caribbean Large Igneous Plateau revealed by 3D data-integrative models"

_Solid Earth, 2020_

## Referee Comment (RC1) · Richard Ernst (Referee) · 4 Nov 2020

Review by R. Ernst of Gómez-García et al. "The preserved plume of the Caribbean Large Igneous Plateau revealed by 3D data-integrative models"

GENERAL COMMENTS:

This is a very exciting manuscript that provides a robust geophysical study and identifies some major lithospheric gravity anomalies that are interpreted to locate two lithospheric fossil plume conduits beneath the Colombian and the Venezuelan basins that locate the head of the former mantle plume inferred responsible for the Caribbean LIP.

[Figure]

There is a wealth of modelling detail, and the paper is extremely written, illustrated and referenced. This also provides a careful consideration of the reconstruction constraints to test whether the Caribbean plume could be linked with the present day Galapagos hotspot, and notes some problems with this linkage based on the most recent reconstructions.

This is an important contribution that should be accepted with only minor corrections.

You focus on the link with the Galapagos hotspot, but I recall the story by Kerr and Tarney (2005 Geology, v. 33; no. 4; p. 269–272; doi: 10.1130/G21109.1), where it was inferred that two plume generated two LIPs that were tectonically amalgamated to form the Caribbean LIP, and that the contributions of the two plumes/LIPs were distinguished on the basis of composition. Could you include a comment on two plume idea of Kerr and Tarney (2005)

One minor suggestion regarding nomenclature: I am considering suggesting that we modify the nomenclature of acronyms for LIPs by introducing a dash between the name the LIP acronym. So the Caribbean LIP would be C-LIP. The rationale is this revised nomenclature is that it would allow us to distinguish different type of LIPs from name of the LIP. Most importantly, we could distinguish silicic LIP (SLIP) with less confusion— e.g. the Whitsunday SLIP would be W-SLIP rather than WSLIP. One could even take this further to distinguish ocean LIP (oceanic plateau) as an OLIP and so the Ontong Java LIP could be OJ-OLIP. Continental LIPs could be also be distinguished as CLIP, and so the Parana-Eteneka LIP could be PE-CLIP. Anyway, for now I wondering about you using C-LIP instead of CLIP for Caribbean LIP.

Detailed comments:

Lines 35: "Although about 12 different oceanic plateaus have been recognised worldwide, they represent one of the least well-known Earth's magmatic processes (Kerr, 2014)." Could you also mention that the older record of oceanic is preserved in orogenic belts since oceanic plateaus are preserved as fragments in orogenic belts during

closure of oceans. (e.g. Dilek, Y. & Ernst, R. (2008). Links between ophiolites and Large Igneous Provinces (LIPs) in Earth history: Introduction. Lithos, 100: 1–13)

Line 58: "Indeed, strong geochemical evidence suggests that the CLIP corresponds to melting of the plume head of the Galápagos hotspot (Geldmacher et al., 2003; Thompson et al., 2004)" As mentioned above could you address the Kerr and Tarney (2005) paper that inferred two different plumes involved in the generation of the Caribbean LIP..

Lines 125-127: "Although there is strong geochemical evidence that supports the origin of the CLIP as melting of the paleo-Galápagos plume head (Geldmacher et al., 2003; Hastie and Kerr, 2010; Kerr and Tarney, 2005; Thompson et al., 2004)," Again could you consider the two plume model of Kerr and Tarney 2005

Line 173: change "slabs shapes" to "slab shapes"— I also note that "flat slab" appears later in the sentence. Could the sentence be edited to reduce the mention of "slab" to one occurrence.

Line 433: "A total volume of ∼12 x 106 km3 is calculated for what is interpreted as the CLIP fossil plume conduits" That is an impressive volume

Lines 448-450: "Previous studies based on geochemical analyses of CLIP-related magmatic rocks have proposed the present-day Galápagos hotspot as the origin of the thermal anomaly responsible for the development of the CLIP (e.g. Duncan and Hargraves, 1984; Geldmacher et al., 2003; Kerr and Tarney, 2005; Pindell and Barrett, 1990; Thompson et al., 2004)." Again consider the two plume model of Kerr and Tarney 2005

Lines 440-441: you mention fossil plume conduits in the Northern East African system. Perhaps you could also mention examples in India, associated with the Deccan LIP and also in the Parana region of Brazil associated with the Parana-Etendeka LIP. Parana –Etendeka LIP (Van Decar et al., 1995), the Deccan LIP (Kennett and Widiyantoro,

1999), and Ontong Java LIP (Richardson et al., 2000; Klosko et al., 2001)

Line 455: change "short after" to "shortly after"

Line 524: change "Particularly" to "In particular"

Line 540: "To summarise, based on the argumentation above, the major offset between the paleo-position of the CLIP at 90 Ma and the present-day Galápagos hotspot can be interpreted either as:. . . . . . . ..2) The CLIP was formed by a different plume, which – if considered fixed - would be nowadays located below the South American continent (Fig. 13)." Consider again the two plume model of Kerr and Tarney 2005

Please also note the supplement to this comment:
https://se.copernicus.org/preprints/se-2020-153/se-2020-153-RC1-supplement.pdf

---

## Author Comment (AC1) · 6 Nov 2020

We would like to thank Richard Ernst for his time and effort in reviewing our contribution. We would like to reply shortly to the main comment regarding the two plumes hypothesis presented by Kerr and Tarney (2005).

It is important to consider that the two plumes are associated with two distinct LIP events. Our work concentrates on the lithosphere of the present-day Caribbean Plate, whereas the second plume hypothesis concerns terranes accreted along the north-western margin of the South American continent (Ecuador and Colombia). This hypothesis has been recently revised by Hochmuth and Gohl (2017) [Hochmuth, K. and

[Figure]

Gohl, K.: Collision of Manihiki Plateau fragments to accretional margins of northern Andes and Antarctic Peninsula, Tectonics, 36(2), 229–240, doi:10.1002/2016TC004333, 201]. Their plate reconstructions suggest that most of these terranes were part of the Manihiki Plateau that formed around 120 Ma and accreted along the NW margin of South America at about 60 Ma. Thus, the magmatic event associated to the second plume (and plateau) doesn't seem to be related to the formation of the present-day Caribbean Plate.

Nevertheless, we will have a look in more details and consider any further implication for our results in the revised version of our manuscript.

---

## Referee Comment (RC2) · Anonymous Referee #2 · 10 Nov 2020

General comments

Gómez-García et al. used 3D data-integrative models to identify lithospheric-scale gravity anomalies (and thus density heterogeneities) across the Caribbean and north-western South American plates. The inversed lithospheric structure is then adopted to reveal the development of the Caribbean Large Igneous Plateau (CLIP) as the fossil plume conduits due to interactions of the Farallon plate and a mantle plume. They presented a comprehensive workflow that utilizes various geophysical datasets to reveal a high-resolution lithospheric structure. That workflow shows great values as it can be easily and widely applied to other areas worldwide. This manuscript is well written

with a clear structure and logic flow, great details of the method and promising results and robust discussions. I highly recommend this work to be published after only minor corrections as showed below.

Lines 269: Can you specify the "medium size wavelength"? Largely in what ranges? Same for the "short wavelength" and "long wavelength" if possible.

Line 272: Further to the previous comment, here can you also give a bit more explanation on how that "medium size wavelength" gravity residual would correspond to density anomalies at depth < 50 km?

Technical corrections

Line 259: Change "... a sharp increase on ..." to "... a sharp increase in ..."

Line 556: Change "...Large Igneous plateaus..." to "...Large Igneous Plateaus..."?

———————————————

---

## Author Comment (AC2) · 10 Nov 2020

Ángela María Gómez-García et al.

**Ángela María Gómez-García et al.**

amgomezgar@unal.edu.co

Dear Anonymous Referee #2,

Thank you very much for reading and commenting on our manuscript. We will consider the minor comments and clarifications you mentioned regarding the wavelengths of the gravity residuals, and will proceed to improve the paper in its second version.

Ángela María Gómez-García (on behalf of the authors).

---

## Author Response (AR1)

**Detailed response to Richard Ernst's comments**

**Comment 1:**

You focus on the link with the Galapagos hotspot, but I recall the story by Kerr and Tarney (2005 Geology, v. 33; no. 4; p. 269–272; doi: 10.1130/G21109.1), where it was inferred that two plume generated two LIPs that were tectonically amalgamated to form the Caribbean LIP, and that the contributions of the two plumes/LIPs were distinguished on the basis of composition. Could you include a comment on two plume idea of Kerr and Tarney (2005).

**Response 1:**

Kerr and Tarney (2005) recognised two different oceanic plateaus in the Caribbean and northwestern South America. However, it is important to point out that they correspond to different tectonic units. While the Caribbean LIP is forming parts of the present day Caribbean plate (below the Caribbean Sea), remnants of a second plateau, are found as accreted terranes along the northwestern margin of the South American continent (see Figure 3 in Kerr and Tarney, 2005).

This hypothesis has been revisited by Hochmuth and Gohl (2017), who suggest that most of these northwestern margin terranes were part of the Manihiki Plateau, which accreted along this margin at about 60 Ma. Indeed, if their reconstructions are correct, the fragments of the Manihiki Plateau should have arrived after the collision between Panama and South American plate (c. 67 Ma –age of development of arc magmatism in Panama -Montes et al., 2012). Therefore, the Manihiki fragments are not expected to be spatially connected with the Caribbean LIP.

Nevertheless, we added a clarifying comment to the text (lines 67-71).

**Comment 2:**

One minor suggestion regarding nomenclature: I am considering suggesting that we modify the nomenclature of acronyms for LIPs by introducing a dash between the name the LIP acronym. So the Caribbean LIP would be C-LIP. The rationale is this revised nomenclature is that it would allow us to distinguish different type of LIPs from name of the LIP. Most importantly, we could distinguish silicic LIP (SLIP) with less confusion— e.g. the Whitsunday SLIP would be W-SLIP rather than WSLIP. One could even take this further to distinguish ocean LIP (oceanic plateau) as an OLIP and so the Ontong Java LIP could be OJ-OLIP. Continental LIPs could be also be distinguished as CLIP, and so the Parana-Eteneka LIP could be PE-CLIP. Anyway, for now I wondering about you using C-LIP instead of CLIP for Caribbean LIP.

**Response 2:**

We modified the text accordingly. Please see tracked version of the manuscript.

**Comment 3:**

Lines 35: "Although about 12 different oceanic plateaus have been recognised worldwide, they represent one of the least well-known Earth's magmatic processes (Kerr, 2014)." Could you also mention that the older record of oceanic is preserved in orogenic belts since oceanic plateaus are preserved as fragments in orogenic belts during closure of oceans. (e.g. Dilek, Y. & Ernst, R. (2008). Links between ophiolites and Large Igneous Provinces (LIPs) in Earth history: Introduction. Lithos, 100: 1–13).

**Response 3:**

We modified the text and added the reference (lines 55-56).

**Comment 4:**

Line 58: "Indeed, strong geochemical evidence suggests that the CLIP corresponds to melting of the plume head of the Galápagos hotspot (Geldmacher et al., 2003; Thompson et al., 2004)" As mentioned above could you address the Kerr and Tarney (2005) paper that inferred two different plumes involved in the generation of the Caribbean LIP.

**Response 4:**

Please refer to Response 1.

**Comment 5:**

Lines 125-127: "Although there is strong geochemical evidence that supports the origin of the CLIP as melting of the paleo-Galápagos plume head (Geldmacher et al., 2003; Hastie and Kerr, 2010; Kerr and Tarney, 2005; Thompson et al., 2004)," Again could you consider the two plume model of Kerr and Tarney 2005.

**Response 5:**

Please refer to Response 1.

**Comment 6:**

Line 173: change "slabs shapes" to "slab shapes"— I also note that "flat slab" appears later in the sentence. Could the sentence be edited to reduce the mention of "slab" to one occurrence.

**Response 6:**

We modified the text accordingly: "6) the shapes of the Nazca (Hayes et al., 2018) and (7) the Caribbean (Mora et al., 2017) flat-slab subductions (Fig. 3 (f))".

**Comment 7:**

Line 433: "A total volume of ~12 x $10^6$ km$^3$ is calculated for what is interpreted as the CLIP fossil plume conduits" That is an impressive volume

**Response 7:**

We double checked the volume calculation and it is correct.

**Comment 8:**

Lines 448-450: "Previous studies based on geochemical analyses of CLIP-related magmatic rocks have proposed the present-day Galápagos hotspot as the origin of the thermal anomaly responsible for the development of the CLIP (e.g. Duncan and Hargraves, 1984; Geldmacher et al., 2003; Kerr and Tarney, 2005; Pindell and Barrett,

1990; Thompson et al., 2004)." Again consider the two plume model of Kerr and Tarney 2005.

**Response 8:**

Please refer to Response 1.

**Comment 9:**

Lines 440-441: you mention fossil plume conduits in the Northern East African system. Perhaps you could also mention examples in India, associated with the Deccan LIP and also in the Parana region of Brazil associated with the Parana-Etendeka LIP. Parana –Etendeka LIP (Van Decar et al., 1995), the Deccan LIP (Kennett and Widiyantoro, 1999), and Ontong Java LIP (Richardson et al., 2000; Klosko et al., 2001).

**Response 9:**

Thank you very much for drawing these manuscripts to our attention. We have included them on lines 456-459.

**Comment 10:**

Line 455: change "short after" to "shortly after". Line 524: change "Particularly" to "In particular"

**Response 10:**

We modified the text accordingly.

**Comment 11:**

Line 540: "To summarise, based on the argumentation above, the major offset between the paleo-position of the CLIP at 90 Ma and the present-day Galápagos hotspot can be interpreted either as: …… 2) The CLIP was formed by a different plume, which – if considered fixed - would be nowadays located below the South American continent (Fig. 13)." Consider again the two plume model of Kerr and Tarney 2005.

**Response 11:**

Please refer to Response 1.

**Detailed response to Reviewer #2 comments**

**Comment 1:**

Lines 269: Can you specify the "medium size wavelength"? Largely in what ranges?
Same for the "short wavelength" and "long wavelength" if possible.

**Response 1:**

We clarified this on the text.

**Comment 2:**

Line 272: Further to the previous comment, here can you also give a bit more explanation on how that "medium size wavelength" gravity residual would correspond to density anomalies at depth < 50 km?

**Response 2:**

We improved the text (lines 278-285).

**Comment 3:**

Technical corrections
Line 259: Change "… a sharp increase on …" to "…a sharp increase in …"
Line 556: Change "…Large Igneous plateaus…" to "…Large Igneous Plateaus…"?

**Response 3:**

Thank you, we modified the text accordingly.

[revised manuscript text omitted]

55  from their mantle plume origin. Indeed, remnants of oceanic plateaus are commonly preserved as accreted fragments within orogenic belts (Dilek and Ernst, 2008)(Dilek and Ernst, 2008). Thus, fFragments of the Caribbean plateau have been accreted along the continental margins, such as of in Ecuador, Colombia, Panama, Costa Rica, Curacao and Hispaniola (Hastie and Kerr, 2010; Thompson et al., 2004). Using accreted material and relatively few drilled or dragged submarine rock samples, the geochemistry of the C-LIPCLIP has been reconstructed (e.g. Geldmacher et al., 2003; Hastie and Kerr, 2010; Kerr and Tarney,

60  2005; Thompson et al., 2004). Indeed, strong geochemical evidence suggests that the C-LIPCLIP corresponds to melting of the plume head of the Galápagos hotspot (Geldmacher et al., 2003; Thompson et al., 2004), although recent kinematic reconstructions of the Caribbean do not allow to trace back the location of the plate above the present-day location of the Galápagos plume (Boschman et al., 2014). Nevertheless, diverse evidence exists about the north - north-eastward migration of the Caribbean plate from the east Pacific. The present-day Caribbean plate is composed of different accreted crustal domains

65  (e.g. volcanic arcs, continental and oceanic realms) which have migrated since Late Jurassic to Early Cretaceous times, including the igneous plateau materials that affected the oceanic crust of the former Farallon plate (Boschman et al., 2014; Montes et al., 2019a). We note that (Kerr and Tarney, (2005) distinguished two plateaus in the Caribbean and northwester South American plates. However, the second plateau correspond to the Manihiki Plateau (Hochmuth and Gohl, 2017), and its fragments collided with the South American continent at about 60 Ma, after the collision between Panama and South American

[revised manuscript text omitted]

**Acknowledgements**

700 The authors thank Richard Ernst and an additional anonymous reviewer for their helpful comments on the manuscript. We would like to thank Prof. Camilo Montes (Universidad del Norte) for the fruitful discussion about the GPlates reconstruction in the Caribbean. AMGG is thankful with Dr. Álvaro González (CRM, Barcelona) for the discussion regarding the tectonic development of the Caribbean. AMGG was partially supported by COLCIENCIAS, Fundación para la Promoción de la Investigación y la Tecnología, CEMARIN, and DAAD research grants.

[revised manuscript text omitted]